# FreeVS: Generative View Synthesis on Free Driving Trajectory

**Qitai Wang**[1,2], **Lue Fan**[2,3], **Yuqi Wang**[1,2], **Yuntao Chen**[4✉], **Zhaoxiang Zhang**[1,2,4✉]

[1]School of Future Technology, University of Chinese Academy of Sciences (UCAS),
[2]NLPR, MAIS, Institute of Automation, Chinese Academy of Sciences (CASIA),
[3]CUHK [4]Center for Artificial Intelligence and Robotics, HKISI, CAS
{wangqitai2020, lue.fan, wangyuqi2020, zhaoxiang.zhang}@ia.ac.cn,
chenyuntao08@gmail.com
Project Page & Code: https://freevs24.github.io/

## ABSTRACT

Existing reconstruction-based novel view synthesis methods for driving scenes focus on synthesizing camera views along the recorded trajectory of the ego vehicle. Their image rendering performance will severely degrade on viewpoints falling out of the recorded trajectory, where camera rays are untrained. We propose FreeVS, a novel fully generative approach that can synthesize camera views on free new trajectories in real driving scenes. To control the generation results to be 3D consistent with the real scenes and accurate in viewpoint pose, we propose the pseudo-image representation of view priors to control the generation process. Viewpoint transformation simulation is applied on pseudo-images to simulate camera movement in each direction. Once trained, FreeVS can be applied to any validation sequences without reconstruction process and synthesis views on novel trajectories. Moreover, we propose two new challenging benchmarks tailored to driving scenes, which are novel camera synthesis and novel trajectory synthesis, emphasizing the freedom of viewpoints. Given that no ground truth images are available on novel trajectories, we also propose to evaluate the consistency of images synthesized on novel trajectories with 3D perception models. Experiments on the Waymo Open Dataset show that FreeVS has a strong image synthesis performance on both the recorded trajectories and novel trajectories. The code is released.

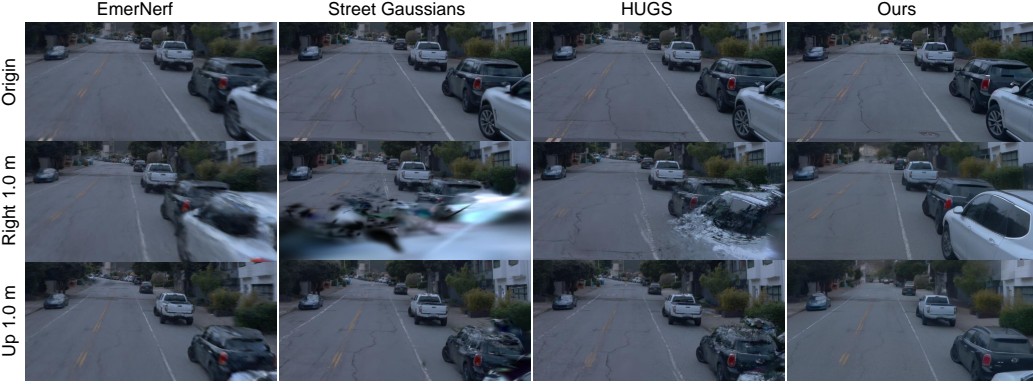

Figure 1: **Synthesis results comparison on the Waymo Open Dataset**(Sun et al., 2020). We show the camera views synthesized by NVS methods on the original front view (first row), viewpoint 1.0 m to the right (second row), and viewpoint 1.0 m above (third tow). Our method significantly outperforms previous NVS methods on viewpoint outside the existing ego trajectory.

# 1 INTRODUCTION

Scene reconstruction and novel view synthesis (NVS) have gained special attention in embodied AI due to their potential to develop closed-loop simulations for embodied systems. Recent advances have led to remarkable improvements in the reconstruction quality of general scenes using multi-pass and multi-view recordings. However, reconstructing driving scenes presents distinct challenges due to the sparse observations inherent in their less controlled, real-world recording conditions.

Unlike general scene reconstruction settings, which typically leverage excessive views surrounding the scene, driving scene reconstruction generally only has access to image views along the single-pass ego driving trajectory. This limitation raises an important question: *How well does driving scene reconstruction perform for novel viewpoints outside the recorded trajectory?*

Currently, existing driving scene NVS works(Guo et al., 2023; Wu et al., 2023c; Xie et al., 2023; Turki et al., 2023; Yang et al., 2023a; Zhou et al., 2024b; Chen et al., 2023; Yan et al., 2024) only evaluate their image rendering quality along the recorded trajectory, leaving this question largely unanswered. As shown in Fig. 1, the quality of rendering results of existing representative NVS methods degrades drastically when the rendering camera moves away from its recording trajectory. This is because, in driving scenes, recorded camera viewpoints are sparse in 3D space and homogeneous in their positions along the recorded trajectories. The sparsity and homogeneity of recorded camera views cause the camera rays shooting from the camera centers on novel trajectories largely untrained.

We propose FreeVS to address this issue, which is a fully generative NVS method that can synthesize high-quality camera views inside and beyond the recorded driving trajectory. We face two core challenges when building the FreeVS . The first challenge is accurately controlling the camera poses while maintaining the 3D geometry consistency of the generated views. Although previous diffusion-based methods(Wang et al., 2023a; Lu et al., 2023b; Hu et al., 2023; Wang et al., 2024b; Yang et al., 2024a) are capable of controlling the camera motion in a coarse trajectory, their control precision is far from enough for safety-critical simulation purposes. The second challenge is the ground truth images in the novel trajectories are unavailable, making it difficult to directly train a model to synthesize novel views beyond recorded trajectories.

To tackle the two challenges, the proposed FreeVS leverages pseudo-image representation, a sparse yet accurate representation of 3D scenes obtained through colored 3D points projection. Specifically, for each existing view, we create its pseudo-image counterpart by projecting colored point clouds into this view. Here the colored points can be easily obtained by projecting point clouds to any valid images. In this way, we obtain training data pairs to train a generative model that can generate a real image from its pseudo-image counterpart. Since we create the pseudo images using ground truth camera models, they contain sparse but highly accurate appearance and geometry, sidestepping the tough challenge of accurately controlling the camera poses. At inference time, we could create a pseudo-image for a novel viewpoint beyond the recorded trajectory and then synthesize the novel view using the trained generative model. This design greatly narrowed the gap between synthesizing views inside and beyond the recorded trajectory.

To reveal the practicality of FreeVS, we propose two challenging benchmarks for evaluating the performance of NVS methods in driving scenes, which is more practically meaningful than the conventional evaluation on the recorded trajectories. (i) On the recorded trajectories, we propose the novel camera synthesis benchmark. Instead of evaluating synthesis results on test frames sampled at intervals from video sequences (i.e. novel frame synthesis), we propose to drop all images of a certain camera view (e.g. the front-side view) in the whole trajectory and synthesize the images of the dropped camera view. (ii) We further propose the novel trajectory synthesis benchmark. With no ground truth images available on novel trajectories, we propose to evaluate the geometry consistency of synthesized views through the performance of off-the-shelf 3D detectors. The experiments on the Waymo Open Dataset (WOD) demonstrate that FreeVS outperforms previous NVS methods by a large margin in the two more practical benchmarks as well as in the traditional novel frame synthesis benchmark.

Our contributions are summarized as follows:

1. We propose FreeVS, a fully generative view synthesis method for driving scenes that generate high-quality 3D-coherent novel views both for recorded and novel trajectories without time-cost reconstruction.

2. We devise two new benchmarks for evaluating driving NVS methods on novel trajectories beyond recorded ones.

3. Experiments on WOD show that FreeVS achieves leading performance on synthesizing camera views inside and beyond the recorded trajectory.

## 2 RELATED WORK

### 2.1 NOVEL VIEW SYNTHESIS THROUGH RECONSTRUCTION

Recently, rapid progress has been achieved in novel view synthesis through 3D reconstruction and radiance field rendering. Neural Radiance Fields (NeRF) (Mildenhall et al., 2020) utilizes multi-layer perceptrons to represent continuous volumetric scenes and achieve a breakthrough in rendering quality. Many works have extended NeRF to unbounded, dynamic urban scenes (Tancik et al., 2022; Barron et al., 2022; Ost et al., 2022; Rematas et al., 2022; Turki et al., 2022; Lu et al., 2023a; Guo et al., 2023; Liu et al., 2023a; Wu et al., 2023c; Xie et al., 2023; Turki et al., 2023; Yang et al., 2023b; Wang et al., 2023b; Ost et al., 2021; Tonderski et al., 2024). Authors of MapNeRF (Wu et al., 2023a) noticed the problem of NeRF in generating extrapolated views and proposed incorporating map priors to guide the training of radiance fields. 3D Gaussian Splatting (Kerbl et al., 2023) (3D GS) models scenes with numerous 3D Gaussians. Recently, some researchers have extended 3D GS to dynamic scenes (Luiten et al., 2024; Wu et al., 2023b; Yang et al., 2023d;c) and driving scenes (Zhou et al., 2024b; Chen et al., 2023; Yan et al., 2024). HUGS (Zhou et al., 2024a) further jointly model the geometry, appearance, motion, and semantics in 3D scenes for better scene understanding.

Another line of work in NVS through reconstruction focuses on fast scene reconstruction or generalizable feed-froward reconstruction(Flynn et al., 2019; Chen et al., 2021; Liu et al., 2022a; Johari et al., 2022; Lin et al., 2022; Varma et al., 2022; Xu et al., 2024; Chen et al., 2025a; Charatan et al., 2024; Wang et al., 2024a; Ren et al., 2024b;a; Zhang et al., 2025; Liu et al., 2025), which are primarily applied to object-centric or small scenes with a few source observations. Recently, SCube(Ren et al., 2024c) and DrivingRecon(?) achieved the feed-forward reconstruction in driving scenes with generalizable models. However, the reconstruction accuracy and rendering resolution of feed-forward reconstruction models in driving scenes still significantly fall short compared to current per-scene reconstruction methods. G3R(Chen et al., 2025b) achieved the fast reconstruction (in a few minutes) of large scenes by efficiently updating a 3D scene representation with generalizable modules that take gradient feedback signals from differentiable rendering as input. Compared to G3R, FreeVS completely does not require a scene reconstruction process, but its image synthesis efficiency is constrained by the efficiency of existing video generation models.

### 2.2 NOVEL VIEW SYNTHESIS THROUGH GENERATION

Novel view synthesis through image generation has greatly benefitted from the advancements in image generation models(Ho et al., 2020; Song et al., 2020; Rombach et al., 2022; Blattmann et al., 2023). Free View Synthesis(Riegler & Koltun, 2020) conditions the image generation process on wrapped image features. Zero-1-to-3(Liu et al., 2023b) and ZeroNVS(Liu et al., 2023b) generate novel views with a diffusion process conditioned on the reference image and the target camera pose embedded as a text embedding. GeNVS(Chan et al., 2023) condition the diffusion process on volume-rendered feature images. Reconfusion(Wu et al., 2024) uses the diffusion model to refine images rendered by the reconstruction model as extra supervision to the reconstruction process. Similarly, RealFusion(Melas-Kyriazi et al., 2023) uses a diffusion model to provide an extra perspective view for object-centric reconstruction. Yu et al. (2024a) use the Stable Video Diffusion model(Blattmann et al., 2023) to iteratively refine the rendered video along a novel camera trajectory based on the partial image wrapped from the reference view to the target view. Most of the previous novel view synthesis through generation works are designed for object-centric(Liu et al., 2023b; Chan et al., 2023; Wu et al., 2024; Melas-Kyriazi et al., 2023; Yu et al., 2024a) or indoor(Liu et al., 2023b; Chan et al., 2023; Wu et al., 2024; Yu et al., 2024a) scenes. MagicDrive3D(Gao et al., 2024) places image generation(Gao et al., 2023) upstream of scene reconstruction to obtain the 3D

representation of an imagined scene. For driving scenes, Yu et al. (2024b) proposes SGD which generates novel views with a diffusion process conditioned on reference images and depth maps in the target view. However, SGD still can only synthesizes camera views along the recorded trajectory of the ego vehicle.

## 3 FREEVS

We introduce the detailed design of our proposed FreeVS in this section. We summarize the algorithm pipeline of FreeVS in Fig. 2.

**Overview of FreeVS.** FreeVS is a fully generative model that synthesizes new camera views on novel trajectories based on observations of 3D scenes from recorded trajectories. FreeVS is implemented as a conditional video diffusion model. To ensure the model generates views from accurate viewpoints with consistent appearance attributes and 3D geometries as the real 3D scene, we formulate all essential priors regarding the 3D scene as pseudo-images to control the diffusion process. Based on view prior conditions, FreeVS is learned by denoising noised target views at training time and synthesizing target views from pure noise at inference time.

### 3.1 VIEW PRIORS FOR VIEW GENERATION

**Unified view prior representation.** One major challenge of generative novel view synthesis is to ensure the generated images are consistent with the priors in the novel view. Here the view priors include the observed colors, 3D geometry, and camera pose of this view. However, the different types of priors are in totally different modalities, posing a significant challenge for diffusion models to precisely encode them. For example, as discussed in Sec. 1, diffusion models cannot precisely control the camera motions (i.e., poses). To tackle this challenge, we propose a pseudo-image representation that unifies all types of view priors in one modality. Pseudo-images are obtained through colored point cloud projection. Specifically, for each frame in a driving sequence, we first merge LiDAR points across the nearby $r$ frames. LiDAR points on moving objects will be merged along the moving trajectory of the object based on their 3D bounding box annotations. Finally, we project the merged and colored LiDAR point cloud into the target camera viewpoints as pseudo-images. In this way, we encode color information, geometry information, and the view pose into a unified pseudo-image, largely facilitating the learning of generative models.

Compared with directly providing reference images and viewpoint transformations to the diffusion process, the pseudo-image representation greatly simplified the optimization objective of the generative model: With the former inputs, the model is required to have a correct understanding of the 3D scene geometry as well as the transformation of viewpoint to generate a correct view based on the reference image. In contrast, with pseudo-image as input, FreeVS only needs to recover target views based on sparse valid pixels, which is more akin to a basic image completion task. The simplification of the training objective greatly enhances the model's robustness to unfamiliar viewpoint transformations, since the generated image is completed from sparse but geometrically accurate pixel points.

**Viewpoint transformation simulation.** Another challenge of novel view generation on new trajectories stems from the absence of ground truth views beyond recorded trajectories. We can only train the generative model on recorded trajectories, where the diversity of viewpoint transformations is extremely limited. For example, we have no access to the training sample where the frontal camera is moved laterally. However, such viewpoint transformation is essential for synthesizing views on novel trajectories at inference time. This brings a significant gap between training and inference for the generative model. Moreover, we propose the viewpoint transformation simulation with pseudo-images. At training time, we sample color and LiDAR priors from frames mismatched with the training image frames. That is to say, we force the generative model to recover current camera views based on observations from nearby frames. Through this, we simulate the camera movement in each direction as a strong data augmentation on pseudo-image priors. For example, as the ego vehicle moves along its heading direction, the side cameras are actually moving to their front or right, Therefore although we have no access to the training data where the front camera is moved laterally, we can still simulate lateral camera movement by training FreeVS on side-views with mismatched observation-supervision frames.

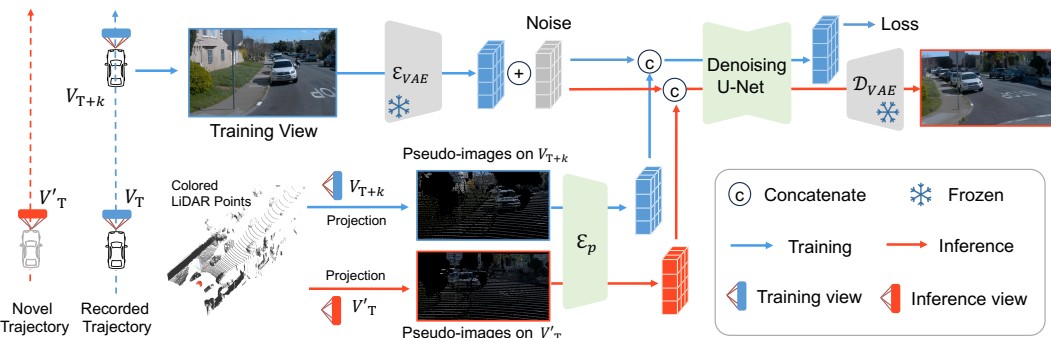

Figure 2: **Method pipeline of FreeVS.** We propose to encode the view priors in driving scenes including appearance, 3D geometry, and pose of target viewpoints all in one modality as the pseudo-images. Best viewed in color. The diffusion model is trained to synthesize target views solely based on the unified pseudo-image priors.

## 3.2 DIFFUSION MODEL FOR NVS

**Training of FreeVS .** In each training iteration of FreeVS , we randomly sample a colored LiDAR point cloud sequence $\mathbf{p} = (p_1, ...p_n)$ from the driving scene dataset. $\mathbf{p}$ is a sequence of colored point clouds, each point cloud frame $p_i \in \mathbb{R}^{N_i \times 6}$ contains a set of 6-dimension 3D points. 3D points are recorded with their positions in the world reference frame and their visible colors. From the driving sequence, we also sample a target camera viewpoint sequence $\mathbf{v} = ([v_1^1, ..., v_1^m], ..., [v_n^1, ..., v_n^m])$ of $n$ frames and $m$ surrounding cameras. Each camera parameter $v_i$ stands for the intrinsics and extrinsics of a camera viewpoint. For a viewpoint in the target video with camera parameter $v_i^j$, we project the colored LiDAR point cloud $p_i$ into the viewpoint as pseudo-image $s_i^j = \text{Proj}(p_i, v_i^j)$. The training target of FreeVS in each iteration is to recover the target images at sampled viewpoints based on the pseudo-image sequence $\mathbf{s} = ([s_1^1, ..., s_1^m], ..., [s_n^1, ..., s_n^m]) \in \mathbb{R}^{n \times m \times 3 \times H \times W}$.

During the training process of FreeVS , the ground truth camera views $\mathbf{x} \in \mathbb{R}^{n \times m \times 3 \times H \times W}$ is also sampled along the viewpoint sequence $\mathbf{v}$. The ground truth camera views are encoded as target video latent representation $\mathcal{E}_{\text{VAE}}(\mathbf{x}) = \mathbf{y} \in \mathbb{R}^{n \times m \times c \times h \times w}$ through an frozen VAE encoder. Then We have the diffused inputs $\mathbf{y}_r = \alpha_\gamma \mathbf{y} + \sigma_\gamma \epsilon, \epsilon \sim \mathcal{N}(\mathbf{0}, \mathbf{I})$, here $\alpha_\tau$ and $\sigma_\tau$ is noise schedule at the diffusion time step $\tau$. We also encode the pseudo-images into the latent representation $\mathcal{E}_p(\mathbf{s}) = \mathbf{z} \in \mathbb{R}^{n \times m \times c \times h \times w}$ with a 2D encoder trained simultaneously with the diffusion model. We concatenate $\mathbf{y}_r$ and $\mathbf{z}$ as the input $\mathbf{k} \in \mathbb{R}^{n \times m \times 2c \times h \times w}$ to the diffusion model to predict the noise upon $\mathbf{y}$. We have a denoising model $\mathbf{f}_\theta$ with parameters $\theta$ that take $\mathbf{y}_r$,$\mathbf{z}$ as inputs and optimized by minimizing the following denoising objective:

$$\mathbb{E}_{\mathbf{k},\tau \sim p_\tau, \epsilon \sim (,)}[\|\epsilon - \mathbf{f}_\theta(\mathbf{k}; \mathbf{c}, \tau)\|_2^2], \tag{1}$$

Where $\mathbf{c}$ is the description conditions generated by encoding the reference camera views with an off-the-shelf CLIP-vision model(Radford et al., 2021), following the convention of diffusion models. $p_\tau$ is a uniform distribution over the diffusion time $\tau$.

**Synthesizing views on novel trajectories with FreeVS .** During the inference process of FreeVS , we project the colored LiDAR points in each frame into the targeted camera poses to generate pseudo-image sequence for image synthesis. The diffusion model is fed with pure noise latents concatenated with pseudo-image latents. The diffused latent will be decoded as synthesized views through an off-the-shelf VAE decoder $\mathcal{D}_{\text{VAE}}$.

## 3.3 EVALUATING NVS ON NOVEL CAMERA AND NOVEL TRAJECTORY SYNTHESIS

To thoroughly demonstrate the view generalization capability of our FreeVS , which can truly meet the demands of closed-loop embodied simulation, we present a comprehensive discussion of evaluation benchmarks for novel view synthesis in driving scenes. Fig. 3 illustrates this: panels (a) and (b) summarize existing evaluation benchmarks, while panels (c) and (d) introduce our two new challenging NVS benchmarks.

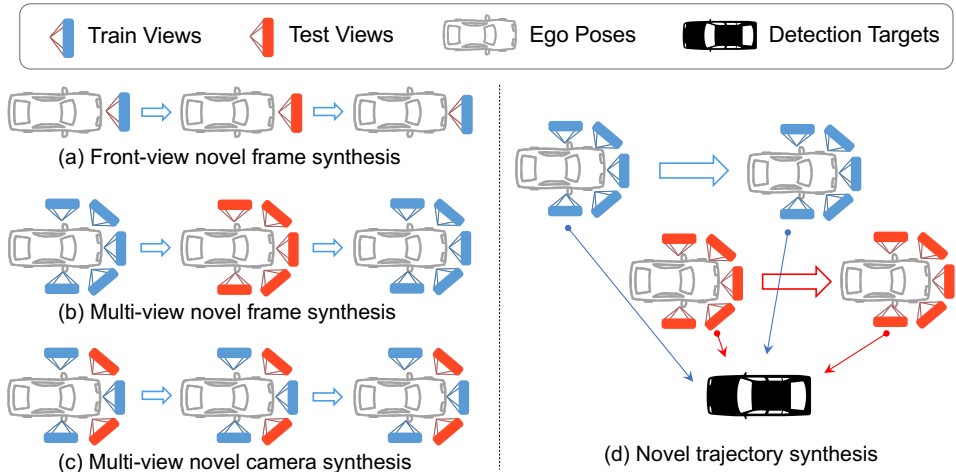

Figure 3: **Benchmarks for evaluating NVS methods in driving scenes.** We conclude the previous NVS evaluation benchmarks for driving scenes as (a) and (b). We propose two novel evaluation benchmarks: the novel camera synthesis benchmark as in (c), and the novel trajectory synthesis benchmark as in (d). Best viewed in color.

**Evaluating NVS on recorded trajectories.** All current NVS works for driving scenes evaluate their NVS performance on test frames sampled periodically along the recorded trajectory. Some previous driving scene NVS methods, such as Street Gaussians(Yan et al., 2024), NSG(Ost et al., 2021), and Mars(Wu et al., 2023c), evaluate their performance with only front camera views considered, as illustrated in Fig. 3(a). Other NVS methods take the multi-view cameras into consideration as illustrated in Fig. 3(b), such as DrivinGaussian(Zhou et al., 2024b), PVG(Chen et al., 2023), EmerN-erf(Yang et al., 2023a), NeuRAD(Tonderski et al., 2024), S-Nerf(Xie et al., 2023), and SUDS(Turki et al., 2023). All these two evaluation benchmarks sample test frames periodically along the trajectory, i.e. **novel frame synthesis**. In such cases, camera views in test frames can be directly inferred from the adjacent frames, especially for datasets with a high video frame rate (e.g. 10Hz for the WOD dataset). To provide a more challenging evaluation setting for driving scene NVS methods, we propose the **novel camera synthesis benchmark** as illustrated in Fig. 3(c). Instead of periodically sampling test frames, we drop images collected by certain multi-view cameras throughout a driving sequence as test views. For example, for a driving sequence in the WOD dataset, we provide NVS methods with the front, and side camera views as training views and evaluate the synthesis results on front-left and front-right views. Under the novel camera synthesis benchmark, NVS methods are required to synthesize views on unseen camera poses, which places higher demands on accurately modeling the 3D scene. We ensure in the validation sequences, most 3D contents in front-side cameras are observed in the front or side camera views along the ego trajectory.

**Novel trajectory synthesis.** On test views sampled from the recorded trajectories, the ground truth camera images are available for evaluating the synthesized images with image similarity metrics including SSIM, PSNR, and LPIPS(Zhang et al., 2018). Differently, in driving scenes, there are no ground truth images available on novel trajectories. The Fréchet Inception Distance (FID)(Seitzer, 2020) metric can compare the overall image distribution between synthesized images on novel trajectories and ground truth images on recorded trajectories, but it can not assess the fidelity of the synthesized images to the 3D scenes at all. In addition to qualitative visualization comparisons, we also propose the **perceptual robustness evaluation** to assess the geometry consistency performance of NVS methods on new ego trajectories.

In driving scenes, modern image-based 3D perception models have achieved high robustness. As shown in Fig. 3(d), assuming an NVS method can synthesize views on a novel trajectory with ideal image quality, the perception model feed with synthesized views should still be able to produce accurate predictions on the novel trajectory. With such an assumption, we believe that the performance of an off-the-shelf perception model on novel trajectories can reflect the quality of images synthesized by the NVS methods. Under the novel trajectory synthesis benchmark, we feed the synthesized images and camera poses on the novel trajectory to an off-the-shelf 3D camera-based

detector. The detection results are evaluated with the longitudinal error tolerant mean average precision (LET-mAP)(Hung et al., 2022) metric on the WOD dataset. For all NVS methods, we modify novel trajectories by laterally shifting the ego positions in each frames. We shift the trajectories by 1.0 m, 2.0 m, and 4.0 m and report the mean evaluated results as $\text{mAP}^{\text{LET}}_{1.0m}$, $\text{mAP}^{\text{LET}}_{2.0m}$, and $\text{mAP}^{\text{LET}}_{4.0m}$.

## 4 EXPERIMENTS

In this section, we first introduce our experimental setup including datasets, evaluation benchmarks, method implementation details, and counterpart methods. Then we provide the quantitative and qualitative experiment results.

**Datasets.** We perform experiments on the WOD dataset(Sun et al., 2020). We select 12 driving sequences for evaluating NVS methods. We ensure that there is ample space on both sides of the ego vehicle in most frames of the selected sequence to simulate novel trajectories by lateral moving the ego vehicle. For each sequence, all 200 data frames in 10Hz are used.

**Evaluation of NVS methods.** We compare FreeVS with NVS counterparts under all the experiment benchmarks shown in Fig. 3. For the front-view or multi-view novel frame synthesis benchmark (Fig. 3(a) and (b)), we sample every fourth frame in driving sequences as test frames. All the remaining frames are used for training NVS counterparts, or as input frames for FreeVS. On, we report metrics including SSIM, PSNR, and LPIPS. Under the novel camera synthesis benchmark, we reserve all the front-side camera views as test views and use the front and side camera views as train views throughout each sequence. Note that for FreeVS which does not require scene reconstruction on validation sequences, we only take information from the train views to generate test views.

Moreover, we also evaluate NVS methods on novel trajectories with the FID score and the proposed perceptual robustness evaluation method. We take MV-FCOS3D++(Wang et al., 2022), a basic yet strong multi-view camera-based 3D detector as our baseline detection model. We follow most of the settings of the official open-sourced implementation of MV-FCOS3D++. We train MV-FCOS3D++ for 24 epochs on the WOD training set (except for the validation sequences in our experiments) to obtain the baseline detector. Following (Wang et al., 2022), we initialize MV-FCOS3D++ from an FCOS3D++ checkpoint, which is also trained on the above training sequences. We feed the camera views synthesized on novel trajectories to the baseline detector. We report camera-based 3D detection metrics LET-mAP(Hung et al., 2022) on the vehicle class as $\text{mAP}^{\text{LET}}$.

**Method details.** We implement the proposed FreeVS pipeline based on Stable Video Diffusion (SVD)(Blattmann et al., 2023). We initialize the diffusion model from a pre-trained Stable Diffusion checkpoint(Rombach et al., 2022). FreeVS is trained on the WOD training set, except for the selected validation sequences. To generate pseudo-images, we accumulate colored LiDAR points across the adjacent $\pm 2$ frames of each frame. If a 3D LiDAR point has more than one projected 2D point in multi-view images, the mean color of its projected image points will be recorded. For viewpoint transformation simulation, we randomly sample the target viewpoint sequence starting from the adjacent $\pm 4$ frames of the first frame of the source point cloud sequence. We employ a ConVNext-T(Liu et al., 2022b) backbone as the pseudo-image encoder. We train the model for 40,000 iterations with a batch size of 8 and video frame length $n = 8$. Please refer to the appendix for more training details.

### 4.1 SOTA COMPARISON UNDER THE PROPOSED CHALLENGING NEW BENCHMARKS.

**Novel camera synthesis.** We first report the performance of NVS methods under the proposed multi-view novel camera synthesis benchmark in Table 1. FreeVS achieves leading performance on all metrics by a large margin. Previous NVS methods tend to render images with severe image distortion or massive unnatural artifacts when facing severe scene information loss on the target views, as shown in Fig. 4. Meanwhile, FreeVS can generate camera views close to ground truth views based on limited 3D scene observations.

**Novel trajectory synthesis.** We also report the FID and perceptual robustness performance of NVS methods on novel trajectories in Table 2. The proposed FreeVS outperforms previous NVS methods on almost all metrics with different lateral offsets applied to the viewpoints. Compared to previous NVS methods, the proposed FreeVS has a very strong performance on the FID metric. This is mainly

Table 1: **Comparison with NVS counterparts on novel camera synthesis.** For all NVS methods, we use all front and side camera views as source views to synthesize the front-side camera views.

| Methods | Front-side camera syntheising | | |
|---|---|---|---|
| | SSIM↑ | PSNR↑ | LPIPS↓ |
| 3D-GS(Kerbl et al., 2023) | 0.484 | 15.97 | 0.442 |
| EmerNerf(Yang et al., 2023a) | 0.603 | 19.61 | 0.330 |
| StreetGaussian(Yan et al., 2024) | 0.531 | 17.35 | 0.397 |
| Ours | **0.628** | **20.70** | **0.283** |

Table 2: **Comparison with NVS counterparts on novel trajectories.** The $y$ axis is defined lateral to the ego vehicle's heading direction. †: performance of baseline detector on ground truth images.

| Methods | $y \pm 0.0m$ | | $y \pm 1.0m$ | | $y \pm 2.0m$ | | $y \pm 4.0m$ | |
|---|---|---|---|---|---|---|---|---|
| | FID↓ | mAP$^{\text{LET}}$↑ | FID↓ | mAP$^{\text{LET}}_{1.0m}$↑ | FID↓ | mAP$^{\text{LET}}_{2.0m}$↑ | FID↓ | mAP$^{\text{LET}}_{4.0m}$↑ |
| GT† | - | 0.895 | - | - | - | - | - | - |
| 3D-GS | 34.79 | 0.729 | 52.07 | 0.605 | 61.16 | 0.581 | 86.21 | 0.452 |
| EmerNerf | 53.88 | 0.600 | 58.26 | 0.510 | 69.50 | 0.478 | 84.81 | 0.464 |
| StreetGaussian | 21.62 | **0.826** | 41.17 | 0.738 | 55.71 | 0.682 | 80.44 | 0.544 |
| Ours | **11.18** | 0.816 | **13.45** | **0.786** | **16.60** | **0.724** | **22.08** | **0.612** |

because the proposed FreeVS is nearly free from image degradation and artifacts when synthesizing images on novel trajectories. FreeVS also has the strongest mAP$^{\text{LET}}$ performance among all NVS methods, which indicates that as a generation-based method, FreeVS is of even higher fidelity to the 3D scene geometry compared with previous reconstruction-based methods when rendering views on novel trajectories. We also provide a visualization comparison example in Fig. 5.

While FreeVS relies on LiDAR point inputs, EmerNerf and Street Gaussians also rely on LiDAR depth supervision during their training process. Therefore FreeVS did not gain any information advantages in our experiments. Moreover, as a fully generative method, FreeVS does not require any scene reconstruction process when applied to validation sequences. From this aspect, at inference time, FreeVS costs less computational resources even compared with 3DGS-based methods, which usually take 1-2 hours to model a validation sequence of 20s.

## 4.2 SOTA COMPARISON ON NOVEL FRAME SYNTHESIS

We also report the performance of NVS methods under the traditional front-view novel frame synthesis or multi-view novel frame synthesis benchmark in Table 3. The performance of previous NVS methods is strong when only the front-view camera is considered. However, when it comes to the multi-view setting which is more aligned with the current autonomous driving scenes, the performance of previous NVS methods is surpassed by the proposed FreeVS by a large margin. It is worth mentioning that in Table 3, all previous reconstruction-based NVS methods exhibit a significant performance drop when multi-view cameras are considered. We think this is due to the increased number of training views, the expanded range of the visible 3D scene, and the rapidly changing content in lateral views, all of which make the convergence of reconstruction models more difficult.

## 4.3 ABLATION STUDIES

**Ablation on view prior condition.** We first ablate on the representation of view prior as conditions for the diffusion process, as shown in Table 4. We apply a breakdown experiment on the pseudo-image representation. Models are trained for 20,000 iterations. We start by dropping the color information in the pseudo-image representation, represented by Table 4(b). Dropping the color nearly does not affect the geometric accuracy of rendered results, but has a considerable impact on the image similarity metrics. Then we experiment with dropping the LiDAR inputs (c), where the reference images and camera pose transformation matrix (from the reference view to the target view) are independently encoded by a VAE or MLP encoder. Under this setting, we found the diffusion model unable to accurately synthesize views on the target viewpoint. Most time, the model ignores the pose condition and moves the camera viewpoint by its familiar viewpoint transformation. (e.g. always move the frontal camera forward or backward, or move the side camera left or right.) Based

Table 3: **Comparison with NVS counterparts on recorded trajectories.** We report the performance of NVS methods when only the front-view cameras are considered or when all multi-view cameras are considered. Reconstruction time cost and FPS are measured under the multi-view setting, with a single NVIDIA L20 GPU.

| Methods | Front View | | | Multi-view | | | Reconstruction time | FPS |
|---|---|---|---|---|---|---|---|---|
| | SSIM↑ | PSNR↑ | LPIPS↓ | SSIM↑ | PSNR↑ | LPIPS↓ | | |
| 3D-GS(Kerbl et al., 2023) | 0.799 | 26.31 | 0.143 | 0.586 | 19.21 | 0.366 | 2-3h† | **61.2** |
| EmerNerf(Yang et al., 2023a) | 0.869 | 30.28 | 0.155 | 0.689 | 24.68 | 0.347 | 2-3h | 0.2 |
| StreetGaussian(Yan et al., 2024) | **0.903** | **30.80** | **0.096** | 0.702 | 22.47 | 0.314 | 1-2h | 52.6 |
| Ours | 0.787 | 25.30 | 0.139 | **0.730** | **24.96** | **0.179** | - | 0.9 |

Table 4: **Ablation study on view prior condition.** We conduct a breakdown ablation on the proposed pseudo-image representation of view priors. The $y$ axis is defined lateral to the ego vehicle's heading direction.

| | View priors | Encoders | Multi-view | | | $y \pm 2.0m$ | |
|---|---|---|---|---|---|---|---|
| | | | SSIM↑ | PSNR↑ | LPIPS↓ | FID↓ | mAP$^{\text{LET}}_{2.0m}$↑ |
| (a) | full priors | 2D-Conv | 0.704 | 23.28 | 0.203 | 21.27 | 0.690 |
| (b) | w/o. color | 2D-Conv | 0.701 | 23.05 | 0.231 | 23.13 | 0.687 |
| (c) | w/o. LiDAR | 2D-Conv + MLP | 0.613 | 19.86 | 0.288 | 21.25 | 0.013 |
| (d) | w/o. projection | 2D-Conv + 3D + MLP | 0.609 | 19.88 | 0.284 | 21.32 | 0.028 |

on (c), we experiment with preserving all view prior inputs but do not unify them as pseudo-images (d). The LiDAR points are encoded as latents with a point cloud backbone(Yan et al., 2018). The experiment result shows the model fails at utilizing LiDAR inputs due to its significant gap with 2D images, the model trained under setting (d) has an identical performance as the mode trained under setting (c). Due to the wrong viewpoint of most generated images, trained models under settings (c) and (d) have extremely poor perceptual robustness performance. Through this observation, we can conclude that the pseudo-image representation greatly improved the overall quality and viewpoint controllability of images generated by the diffusion model.

**Ablation study on viewpoint transformation simulation.** We report the results of ablation studies on viewpoint transformation simulation with pseudo-images in Table 5. We report the performance of FreeVS under the multi-view novel frame synthesis setting and on novel trajectories generated by applying 2.0 m lateral offsets to the recorded trajectories. As shown in Table 5, sampling target frames from adjacent $\pm 2$ or $\pm 4$ frames from the source frame can boost the view-synthesize performance of FreeVS on novel trajectories. When the temporal sampling window size exceeds $\pm 4$ frames, the view-synthesize performance of FreeVS on the recorded trajectory will be negatively affected. We believe this is due to the large timestamp mismatch between view priors and target images hindering the model's convergence. We also present a visualization illustration of the effect of viewpoint transformation simulation in the appendix.

## 4.4 VISUALIZATION COMPARISON

We show visualization comparisons between NVS methods under the novel camera synthesis benchmark in Fig. 4, and under the novel trajectory synthesis benchmark in Fig. 5.

## 4.5 CONCLUSION

We present FreeVS, a novel fully generative method for synthesizing camera views on free driving trajectory. We propose the pseudo-image representation of view priors, which conveys accurate 3D scene geometry and viewpoint conditions through colored point projection. The diffusion model is trained to synthesize target views solely based on pseudo-images. In this paper, we fully discuss the evaluation benchmarks for driving scene NVS. We propose two novel evaluation benchmarks including the novel camera synthesis benchmark and the novel trajectory synthesis benchmark. We also propose the perceptual robustness evaluation method for assessing the performance of NVS methods on novel trajectories. Experiments across several experiment benchmarks show that FreeVS achieves leading performance in synthesizing camera views inside or beyond recorded trajectories.

Table 5: **Ablation study on viewpoint translation simulation.** The $y$ axis is defined lateral to the ego vehicle's heading direction.

| Temporal sampling window size | Multi-view | | | $y \pm 2.0m$ | |
|---|---|---|---|---|---|
| | SSIM↑ | PSNR↑ | LPIPS↓ | FID↓ | mAP$_{2.0m}^{LET}$↑ |
| - | 0.733 | 25.04 | 0.180 | 16.93 | 0.707 |
| $\pm2$ frames | 0.734 | 25.04 | 0.179 | 16.77 | 0.713 |
| $\pm4$ frames | 0.730 | 24.96 | 0.179 | 16.60 | 0.724 |
| $\pm8$ frames | 0.717 | 24.82 | 0.188 | 16.53 | 0.721 |

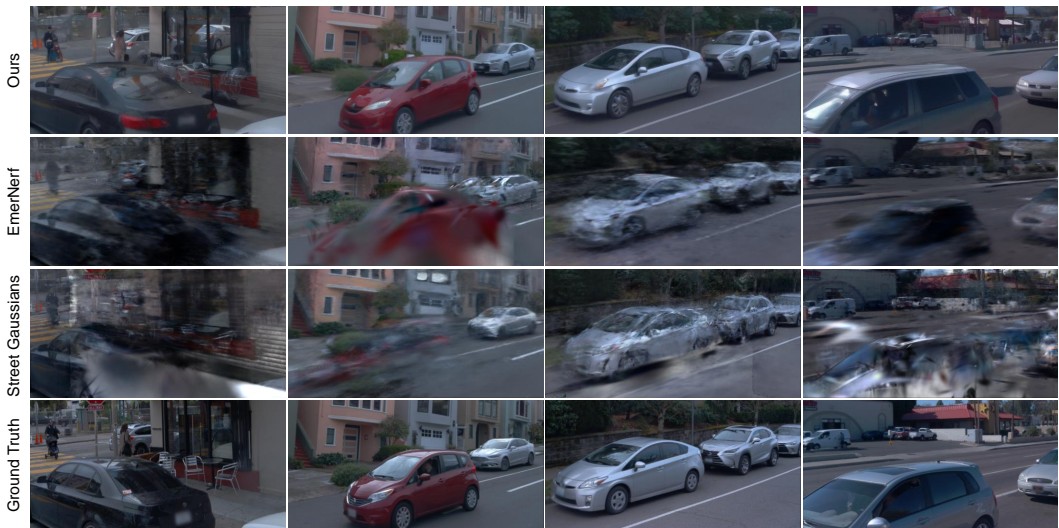

Figure 4: **Visualization comparison on novel-camera synthesis benchmark.** We show the front-side camera views synthesized from front and side camera views with NVS methods.

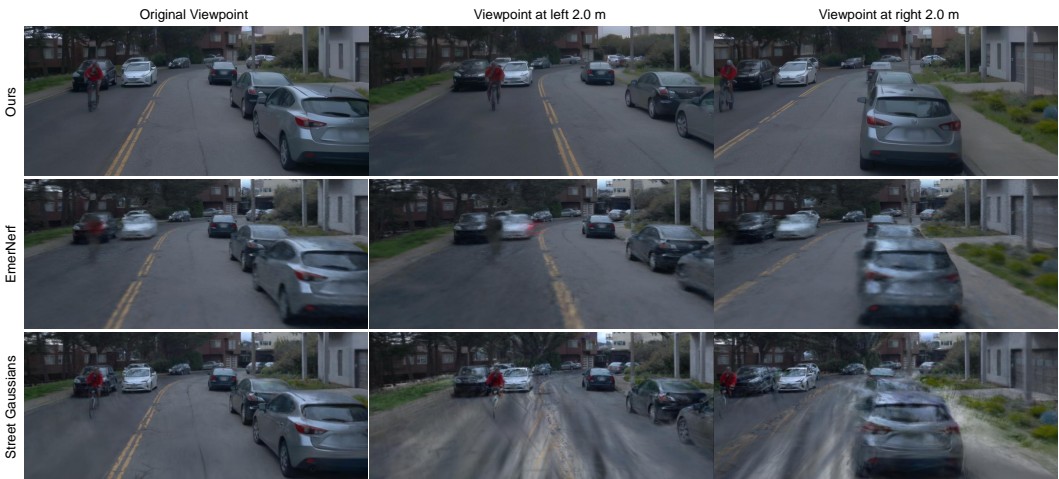

Figure 5: **Visualization comparison on novel trajectories.** We show the camera views synthesized by NVS methods on the original training viewpoint, viewpoint 2.0 m left of the original viewpoint, and viewpoint 2.0 m right of the original viewpoint.

ACKNOWLEDGMENTS

This work was supported in part by the National Key R&D Program of China (No. 2022ZD0116500), the National Natural Science Foundation of China (No. U21B2042, No. 62320106010), and in part by the 2035 Innovation Program of CAS.

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

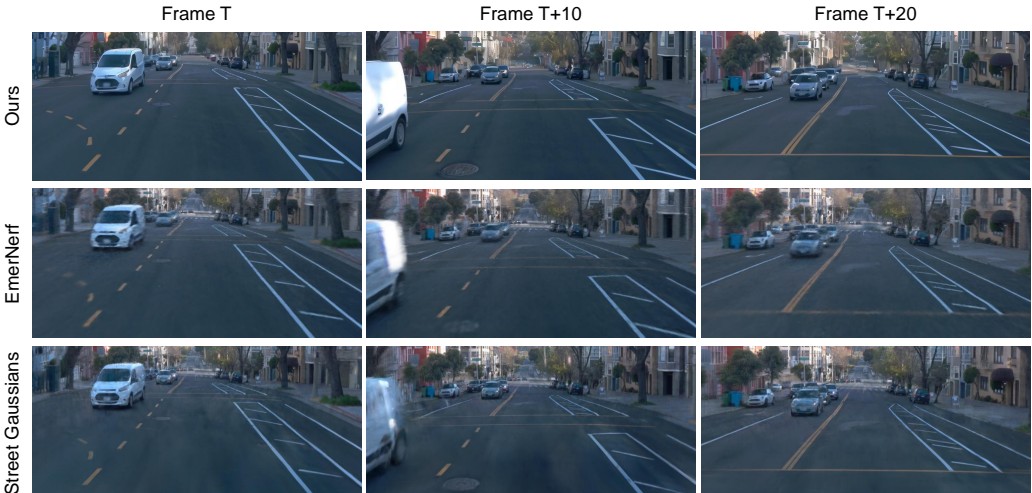

Figure A: **Visualization comparison on moving objects.** We compare the performance of NVS methods where moving objects are visible in camera views. In the shown case, our proposed method can generate more accurate images of vehicles driving in the opposite lane. Images are synthesized on training views.

## A  APPENDIX

### A.1  IMPLEMENTATION DETAILS

**Implementation details of FreeVS.** We employ a ConVNext-T(Liu et al., 2022b) encoder to encode pseudo-images. For training FreeVS, the diffusion model is initialized with Stable Diffusion checkpoints (Rombach et al., 2022). We train the model for 40,000 iterations with a batch size of 8 and video frame length $n = 8$. We use the AdamW optimizer (Kingma & Ba, 2014) with a learning rate $1 \times 10^{-4}$. During training time, we randomly drop the pseudo-image condition latent as well as the CLIP text description latent with a probability of 20%. We enable the viewpoint transformation simulation with a probability of 50%. During inference, we set the number of sampling steps as 25 and stochasticity $\eta$=1.0. When synthesizing images on the existing trajectory, we set the classifier-free guidance (CFG)(Ho & Salimans, 2022) guidance scale to 1.0. For synthesizing images on novel cameras and new trajectories, we enlarge the CFG guidance scale to 2.0 to strengthen the control of 3D prior conditions over the generated results.

**Implementation details of NVS counterparts.** We compare our novel view synthesis method with the 3DGS(Kerbl et al., 2023), EmerNeRF(Yang et al., 2023a), and Street Gaussians(Yan et al., 2024). All counterpart methods are implemented based on their official implementation. For 3DGS which is not initially designed for unbounded driving scenes, we largely increase its max training iterations for better convergence of the model. Please check the appendix for more implementation details on NVS counterparts. For 3DGS which is not initially designed for unbounded driving scenes, we optimize its performance by adjusting its hyperparameters, including setting the densification interval to 500 iterations, setting the opacity reset interval to 10000, and training the 3DGS models for 100000 iterations while densifying 3D Gaussians until 50000 iterations. We also noticed that the Street Gaussians models have a convergence issue when training with all 200 frames of each sequence. Therefore, we split the validation sequences into two 100-frame sequences for training the Street Gaussians models, following its official configuration(Yan et al., 2024). We resize the input images on WOD from a resolution of $1920 \times 1280$ to $1248 \times 832$ and clipped the sky region in images (top 40% area in front views and top 13% area in side views) when training FreeVS and all NVS counterparts. The pseudo-images are generated in the same resolution. All experiments are conducted on NVIDIA L20 GPUs.

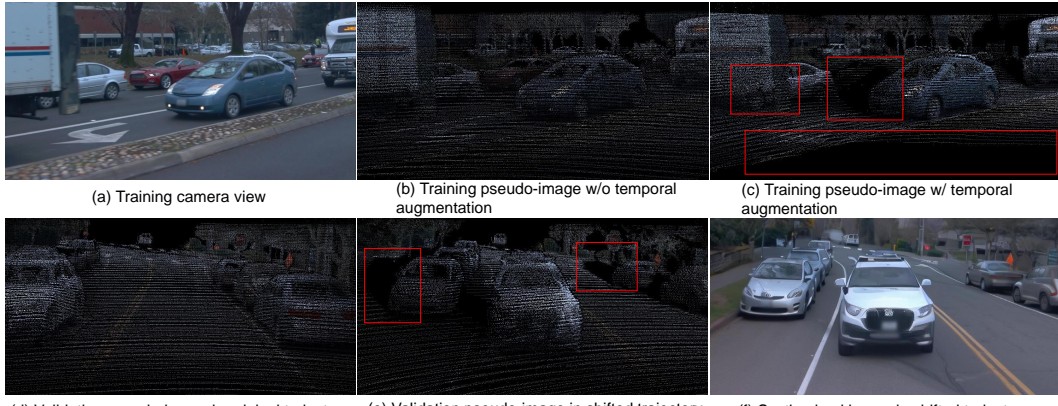

Figure B: **Visualization of the viewpoint transformation simulation.** For a training sample, we show the (a) source camera view, (b) pseudo-image generated from current LiDAR observation, and (c) pseudo-image generated from previous LiDAR observation to simulate viewpoint transformation. We also show a validation case of (d) pseudo-image on the original viewpoint, (e) pseudo-image on the shifted viewpoint, and (f) generated image on the shifted viewpoint. Note the image areas with missing or overlapped 3D observation circled by red boxes in (c) and (f). The proposed viewpoint transformation simulation on pseudo-images can well-stimulate the insufficient 3D prior observations brought by the shift in viewpoint.

## A.2   VIDEO COMPARISON.

We present a video comparison of novel view sequences synthesized by NVS methods on the modified trajectory in a driving sequence. Please check the video file submitted as supplementary material.

## A.3   VISUALIATION COMPARISON ON DYNAMIC OBJECTS.

Without the scene reconstruction process, FreeVS is free from complex cross-frame optimization of dynamic objects. FreeVS can synthesize images of dynamic objects in high accuracy, as shown in Fig.A. In comparison, despite with specific design, current reconstruction-based NVS methods still suffer from dynamic object modeling.

## A.4   VISUALIZATION OF THE VIEWPOINT TRANSFORMATION SIMULATION.

Besides quantitatively ablating the impact of the proposed viewpoint transformation simulation with pseudo-images, we also provide a visualization case in Fig.B to illustrate its effect.

When facing the translation of viewpoints, pseudo images generated from the existing viewpoints cannot provide complete 3D priors, such as in areas circled by red boxes in Fig.B(e). To strengthen the robustness of FreeVS towards those patterns in pseudo-images, we propose to employ viewpoint transformation simulation with pseudo-images by generating pseudo-images in training views from mismatched LiDAR observations. As shown in Fig.B(c), we can stimulate the pseudo-image areas with insufficient 3D priors. The generative model trained with the proposed viewpoint transformation simulation with pseudo-images can render images of high quality when facing insufficient 3D prior inputs, as shown in Fig.B(f).

## A.5   VISUALIZATION RESULTS OF COMPONENT ABLATION EXPERIMENT ON PSEUDO-IMAGE REPRESENTATION.

For the ablation setting (a) and (d) introduced in Sec.4.3, whose quantitative results are reported in Table 4, we present a visualization comparison in Fig.C. Setting (a) is the baseline setting of feeding diffusion models with pseudo-image scene representations for view synthesis. Under setting (d), we feed the diffusion model with the reference image, LiDAR point cloud, and the transformation

Pose encoded in pseudo-image          Pose as transformation matrix

Figure C: **Visualization comparison on the encoding of camera viewpoint condition.** We qualitatively compare the image synthesis performance of the diffusion model with pseudo-image as input (Table 4(a)), or with the reference image, LiDAR points, and viewpoint transformation matrix as input (Table 4(d)). When feeding the model with pose transformation matrices, the diffusion models often fail to generate views on the targeted viewpoint, as shown in the second column.

Table A: **Comparison with UC-NeRF on recorded trajectories.** We report the performance of FreeVS, UC-NeRF, and EmerNerf when the three frontal view cameras are considered.

| Methods | SSIM↑ | PSNR↑ | LPIPS↓ |
|---|---|---|---|
| EmerNerf(Yang et al., 2023a) | 0.764 | 25.16 | 0.282 |
| UC-NeRF(Cheng et al., 2023) | 0.770 | 25.91 | 0.250 |
| Ours | 0.761 | 25.47 | 0.146 |

matrix from the reference view to the target view. As shown in Fig.C, model feed with pseudo-image can precisely synthesize image on the target viewpoint, while model feed with raw camera pose fails to follow the viewpoint condition. Given that the diffusion model can only be trained on recorded trajectories, we found the diffusion model fed with reference images and viewpoint pose tends to overfit to the specific camera movement pattern in each camera position. As shown in Fig.C, the model fed with raw 3D prior inputs will only move the frontal camera view forward or backward, ignoring the viewpoint pose condition. This is due to the absence of a training sample where the frontal camera is moved laterally. By modifying the novel view synthesis task as an image completion task based on the pseudo-image representation of 3D priors as well as applying the proposed viewpoint transformation simulation with pseudo-images, we can overcome this training data shortage.

### A.6 COMPARE WITH METHODS WITH IMAGE WARPING

The baseline methods in the main paper, such as EmerNeRF and StreetGaussian, do not possess special designs for rendering views out of the recorded trajectories. On the other hand, some previous works including UC-NeRF(Cheng et al., 2023) and HO-Gaussian(Li et al., 2024b) use image wrapping based on predicted depth maps to strengthen their reconstructed results. Such design might alleviate the overfitting problem of reconstruction methods on recorded trajectories, although they are not designed to strengthen the performance of those methods on new trajectories. Here we report the performance of UC-NeRF on our validation segments as a supplement to the experiment results in the main paper. We only consider the three frontal cameras in the following supplementary experiments, due to UC-NeRF's mostly hard-coded implementations and hyper-parameters optimized for the 3-camera setting. We compare FreeVS with UC-NeRF under the novel-frame synthesis setting on the recorded trajectories and under the novel-trajectory synthesis setting. Considering that dropping the side cameras will have some impact on the overall performance of each method, we

Table B: **Comparison with UC-NeRF on new trajectories.** We report the performance of FreeVS, UC-NeRF, and EmerNerf when the three frontal view cameras are considered.

| Methods | $y \pm 1.0m$ | | $y \pm 2.0m$ | | $y \pm 4.0m$ | |
| | FID↓ | mAP$_{1.0m}^{LET}$↑ | FID↓ | mAP$_{2.0m}^{LET}$↑ | FID↓ | mAP$_{4.0m}^{LET}$↑ |
| --- | --- | --- | --- | --- | --- | --- |
| EmerNerf | 50.58 | 0.541 | 62.86 | 0.506 | 79.61 | 0.418 |
| UC-NeRF | 42.08 | 0.592 | 75.56 | 0.447 | 88.47 | 0.370 |
| Ours | 15.17 | 0.781 | 18.49 | 0.761 | 24.07 | 0.696 |

Table C: **Performance on the nuScenes dataset.** The $y$ axis is defined lateral to the ego vehicle's heading direction.

| Methods | $y \pm 0.0m$ FID↓ | $y \pm 2.0m$ FID↓ | $y \pm 4.0m$ FID↓ |
| --- | --- | --- | --- |
| PVG | 65.08 | 87.65 | 113.41 |
| Ours | 35.32 | 39.49 | 43.31 |

also provide the performance of EmerNerf under the 3-camera setting as a reference. We report the performance of NVS methods on the recorded trajectories in Table A, and on new trajectories in Table B. On the recorded trajectories, UC-NeRF and FreeVS have similar performance (UC-NeRF has a slightly better SSIM/PSNR performance, while FreeVS has a significantly better LPIPS performance.) On the new trajectories, FreeVS significantly outperforms UC-NeRF.

## A.7 EXPERIMENT ON THE NUSCENES DATASET

We report the performance of FreeVS on the nuScenes dataset in Table C. We randomly select 8 sequences from the nuScenes dataset for experiments. We also report the performance of PVG(Chen et al., 2023) following its unofficial implementation[1] on the nuScenes dataset. We resize the input images to half of the original resolution ($800 \times 450$) when training FreeVS. PVG models are trained under full-resolution image inputs. On the nuScenes dataset, FreeVS still demonstrates the ability to synthesize high-quality images on new trajectories. We also show the novel trajectory synthesis examples of FreeVS on the nuScenes dataset in Fig.D

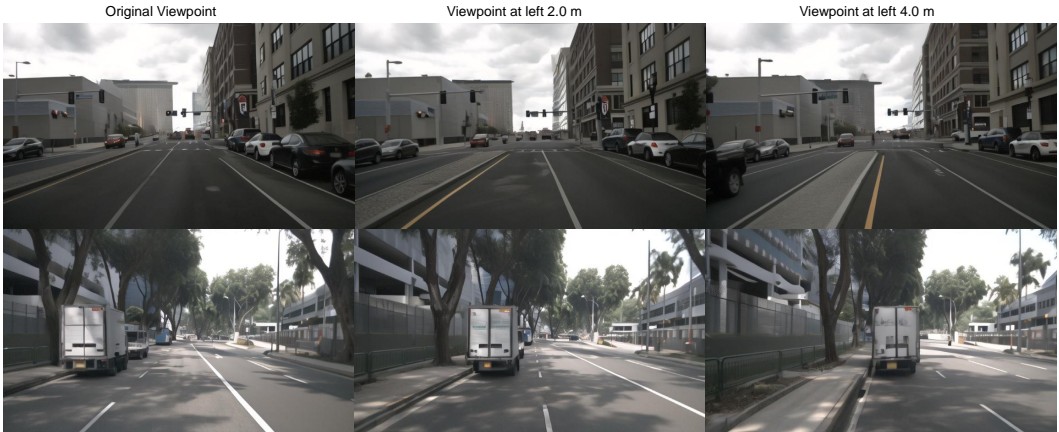

Figure D: **Qualitative results on the nuScenes dataset.**

Table D: **Comparison with NVS counterparts on the XLD dataset.** We report the performance of previous NVS methods according to Li et al. (2024a).

| Methods | $y + 4.0m$ | | | $y + 2.0m$ | | | $y + 1.0m$ | | | $y + 0.0m$ | | |
|---|---|---|---|---|---|---|---|---|---|---|---|---|
| | PSNR↑ | SSIM↑ | LPIPS↓ | PSNR↑ | SSIM↑ | LPIPS↓ | PSNR↑ | SSIM↑ | LPIPS↓ | PSNR↑ | SSIM↑ | LPIPS↓ |
| UC-NeRF | 22.89 | 0.768 | 0.420 | 25.17 | 0.863 | 0.367 | 30.07 | 0.896 | 0.355 | 35.95 | 0.936 | 0.311 |
| MARS | 23.29 | 0.818 | 0.235 | 24.95 | 0.847 | 0.194 | 27.40 | 0.851 | 0.169 | 30.21 | 0.873 | 0.146 |
| EmerNeRF | 24.80 | 0.837 | 0.203 | 26.05 | 0.852 | 0.182 | 28.66 | 0.878 | 0.150 | 31.76 | 0.907 | 0.126 |
| PVG | 23.17 | 0.841 | 0.353 | 24.42 | 0.854 | 0.335 | 26.84 | 0.882 | 0.296 | 37.78 | 0.960 | 0.189 |
| Ours | 26.52 | 0.865 | 0.151 | 27.65 | 0.876 | 0.135 | 29.08 | 0.894 | 0.120 | 30.06 | 0.908 | 0.079 |

## A.8 EXPERIMENT ON THE XLD DATASET

The XLD dataset(Li et al., 2024a) is a synthesized dataset with ground-truth camera views on trajectories with 1.0m, 2.0m, or 4.0m offset from the training trajectories. On the XLD dataset, we can evaluate the performance of FreeVS with image-recovering metrics including PSNR, SSIM, and LPIPIS. Here we report the performance of FreeVS on XLD dataset on trajectories with 0.0m, 1.0m, 2.0m or 4.0m offsets from the training trajectories in Table D.

Experiments show that FreeVS outperforms all previous methods on all metrics on trajectories with 2m or 4m offsets. On trajectories with 1m offsets, the performance of FreeVS is still better than most previous methods. On trajectories with no offset, reconstruction-based methods have better PSNR and SSIM performance. Still, we think the high performance of reconstruction-based methods on the original trajectories comes from their overfitting on the training views, considering the huge performance gap between their performance on trajectories with no offset and trajectories with a slight 1m offset (-5.88 PSNR for UC-NeRF,-10.94 PSNR for PVG).

It is also worth emphasizing that FreeVS has a significantly better LPIPS performance on all validation trajectories, even including trajectories with no offsets. As a perceptual metric, LPIPS is more aligned with human perception compared with pixel-error metrics such as PSNR/SSIM. As the performance degradation of the reconstruction-based methods mainly comes from artifacts in the image when facing out-of-domain test views, the performance degradation of FreeVS mainly comes from the loss of high-frequency details such as the brick patterns on a wall or the number of leaves on a tree. We believe this is why FreeVS exhibits significantly better visual appeal in visualized results.

We also present a qualitative comparison between FreeVS and reconstruction-based methods in Figure F. We compare the generation results of FreeVS with all visualization results shown in Figure 3 in the XLD dataset paper(Li et al., 2024a).

## A.9 LIMITATION DISCUSSION

As discussed above, the pseudo-image representation of 3D priors can handle rigid dynamic objects like vehicles well by accumulating their 3D points along their moving trajectory. On the other hand, the pseudo-image representation cannot handle non-rigid dynamic objects (like pedestrians and cyclists) as well as rigid objects since their 3D points are hard to be correctly accumulated across frames. However, thanks to the generalization ability of video generation models, views of pedestrians and cyclists synthesized by FreeVS are still satisfying as shown in Figure 5 in the paper.

As we use ground truth LiDAR points for the main experiments, we found the LiDAR inputs for FreeVS can be replaced by pseudo-LiDAR points generated from estimated depths. We experimented with applying an off-the-shelf Depth-Anything-V2(Yang et al., 2024b) model on reference images and sample pseudo LiDAR points from the obtained depth map. Here we show a success case as well as a failure case of generation with pseudo LiDAR points in Figure **??**. According to our observation, the proposed FreeVS pipeline can be applied to pseudo LiDAR points and recover most scene contents correctly. However, we found the depth prediction model sometime has problems in predicting the depth of objects very near to the camera, as shown in the failure case, circled with red. We believe this issue is caused by directly applying the Depth-Anything-V2 model to the WOD dataset. A depth predictor that is fully converged on the WOD dataset should be able to avoid this problem. As for FreeVS, the visualized results show that our method has satisfying performance on most scene contents with about the right depth.

---

[1]https://github.com/ziyc/drivestudio/

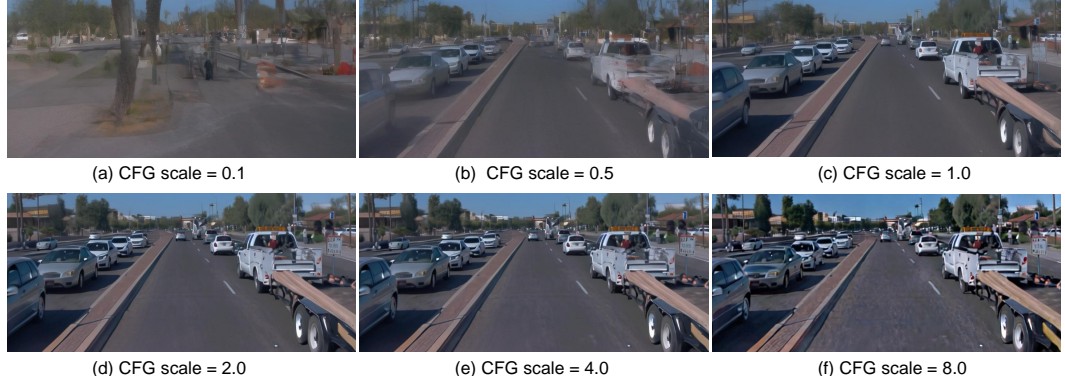

|  |  |  |
|---|---|---|
| (a) CFG scale = 0.1 | (b) CFG scale = 0.5 | (c) CFG scale = 1.0 |
| (d) CFG scale = 2.0 | (e) CFG scale = 4.0 | (f) CFG scale = 8.0 |

Figure E: **Visualization of the effect of classifier-free guidance (CFG).** We show generation results with the same input pseudo-image and different CFG scales. The larger CFG scale is set, the stronger the generation result is constrained by the 3D prior condition.

Note that we generate the above-provided results by directly applying the FreeVS model trained with GT LiDAR points on the pseudo LiDAR points. The different valid pixel patterns and pixel density of pseudo image generated from pseudo LiDAR points lead to some slight imaging style changes of the generated result, which is totally avoidable by training FreeVS on pseudo image generated from pseudo LiDAR points.

### A.10 EFFECT OF CLASSIFIER-FREE GUIDANCE

As a diffusion model, FreeVS can use the classifier-free guidance (CFG) technique to adjust the control effect of input 3D prior condtions. We show the impact of CFG with different CFG scales in Fig. E.

### A.11 VALIDATION SEQUENCES

We list the selected 12 validation sequences from the WOD dataset here with their official individual file names:

- segment-10588771936253546636_2300_000_2320_000_with_camera_labels.tfrecord,
- segment-6242822583398487496_73_000_93_000_with_camera_labels.tfrecord,
- segment-16801666784196221098_2480_000_2500_000_with_camera_labels.tfrecord,
- segment-11917887760630624072_3880_000_3900_000_with_camera_labels.tfrecord,
- segment-10625026498155904401_200_000_220_000_with_camera_labels.tfrecord,
- segment-11846396154240966170_3540_000_3560_000_with_camera_labels.tfrecord,
- segment-18111897798871103675_320_000_340_000_with_camera_labels.tfrecord,
- segment-11017034898130016754_697_830_717_830_with_camera_labels.tfrecord,
- segment-10963653239323173269_1924_000_1944_000_with_camera_labels.tfrecord,
- segment-12161824480686739258_1813_380_1833_380_with_camera_labels.tfrecord,
- segment-11928449532664718059_1200_000_1220_000_with_camera_labels.tfrecord,
- segment-10444454289801298640_4360_000_4380_000_with_camera_labels.tfrecord.

### A.12 EFFICIENCY COMPARISON WITH GENERALIZABLE RECONSTRUCTION METHODS.

Here we provide an efficiency comparison between FreeVS and generalizable 3D reconstruction methods in Table E. Note that we report the performance of generalizable 3D reconstruction methods in the PandaSet dataset according to (Chen et al., 2025b), and the performance of FreeVS in the WOD dataset. The PandaSet dataset consists of video sequences in 80 frames (on average) and 6

cameras, while WOD dataset consists of video sequences in 200 frames and 5 cameras, therefore the comparison experiment setting is not completely aligned. The 3D scale of scenes in PandaSet or WOD seems similar. The training time cost, reconstruction time cost, and rendering FPS are measured on RTX 3090 GPU according to (Chen et al., 2025b), and the computational costs of FreeVS are measured on a single NVIDIA L20 GPU.

Table E: **Efficiency comparison with generalizable reconstruction methods.** †: methods that need to reconstruct the scene again with different source images when rendering each new view. *: The training costs of generalizable reconstruction methods are measured on 2 RTX 3090 GPUs, while the training cost of FreeVS is measured on 8 NVIDIA L20 GPUs. Samely, the inference efficiency of previous methods / FreeVS is measured on 3090 / L20 GPU.

| Methods | Train Time | Train mem | Recon mem | Recon time | Render FPS |
|---|---|---|---|---|---|
| ENeRF(Lin et al., 2022) | 108h*2 | 24GB | 10GB | 0.11s† | 2.65 |
| GNT(Varma et al., 2022) | 49h*2 | 23GB | 21GB | 0.35s† | 0.00249 |
| PixelSplat(Charatan et al., 2024) | 110h*2 | 48GB | 11GB | 1.14s † | 176 |
| G3R(Chen et al., 2025b) | 60h*2 | 20GB | 24GB | 210s | 97.0 |
| FreeVS | 40h*8 | 45GB | 14GB | 0s | 0.9 |

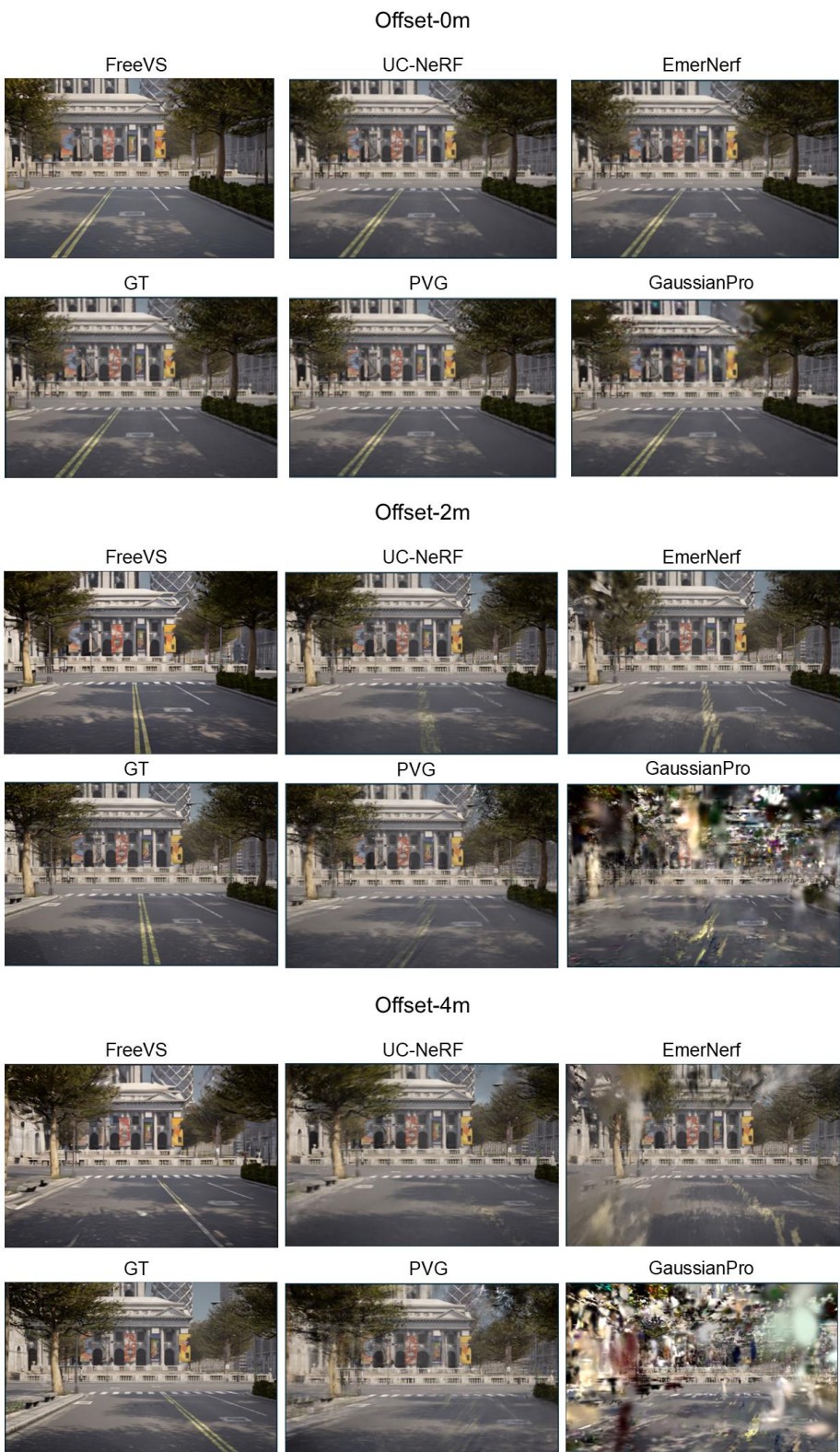

Figure F: **Visualization comparison with NVS counterparts on the XLD dataset.** We qualitatively compare the image synthesis performance of FreeVS with previous NVS methods on the XLD dataset. Visualization results of previous NVS methods are referenced from Li et al. (2024a).

