# OpenReview forum: "FreeVS: Generative View Synthesis on Free Driving Trajectory"
_ICLR.cc/2025/Conference — ICLR 2025 Poster_

### Official Review · Reviewer_BdtD · 2024-10-25

**Soundness:** 3
**Presentation:** 3
**Contribution:** 2
**Rating:** 3
**Confidence:** 4

**Summary:**

This paper addresses the free-view synthesis problem by training a diffusion model conditioned on synthetic images through point cloud projection. During training, it project colored LiDAR points from nearby views into the reference view as a condition image, and finetune a diffusion model so that it can generate the reference view. During inference, condition image can be synthesized for a novel view, which is taken by the diffusion model to generate a realistic image of that view.

**Strengths:**

The paper is easy to read and the idea is clear.

The method is solid and reasonable, potentially showing good generalizability without training very big models (e.g. video foundation model). Conditioning on pseudo-images can help improve 3D consistency.

**Weaknesses:**

1. the proposed method seems too normal, and I read tons of paper doing similar things -- fine-tune the result based on a synthesized image. The authors also mentioned several such works in the related work. The only difference seems to be that they deal with objects while the authors deal with the driving scene.

2. it seems that the paper did not seriously consider temporal information. There is no way to ensure temporal consistency. Also, it seems that moving objects can cause inconsistency and troubles during pseudo image synthesis while the authors only have one sentence discussing how to address it. I believe it can cause a lot of troubles.

3. There is no discussion about the requirement of the pseudo images. For examples, if the novel view is far from captured views, it surely can cause problems.

**Questions:**

See weakness.

---

> ### Author Response · Authors · 2024-11-17
> **Response to Reviewer BdtD (1/3)**
>
> Thank you for your valuable concerns and suggestions.
>
> **First of all, we would like to provide several long video sequences generated by FreeVS on an anonymized page as supplements to the submitted video demo:** https://anonymize289.github.io/anonymized289.github.io/
>
> Below are our responses to your comments:
>
> **W1: About the proposed method seems too normal**
>
> As discussed and acknowledged by many previous works [1-9], novel view synthesis (NVS) in driving scenes requires solutions significantly different from NVS methods for object-centric scenes.
>
> Since no specific works are referenced in the review comment,  here we generally list the major differences between NVS for  object-centric scenes and NVS for driving scenes:
>
> 1. **Narrow, static scenes vs. open, dynamic scenes.** Object-centric scenes usually only involve a single, static object in a narrow scene, while driving scenes are unbounded scenes with rich dynamic objects.
> 2. **Dense, rich viewpoints vs. sparse, trajectory-bounded viewpoints.**  In object-centric scenarios, the camera views are distributed surrounding an object. In driving scenes, camera views are sparsely and homogeneously distributed along the recorded trajectory.  This makes training a model to synthesize views beyond the recorded trajectories a key challenge. Tackling this challenge is one of the main goals of our paper.
> 3. **Synthesizing monocular images vs. synthesizing multi-view videos.** Views in driving scenes are usually required to be synthesized as multi-view videos, while views in object-centric scenes are usually monocular images with no temporal relationship.  This places higher demands on the temporal and spatial consistency of view synthesis in driving scenes.
>
> The proposed generation pipeline, with pseudo-image as 3D priors, is specially designed to address the above-listed challenges in driving scenes：
>
> 1. **Dynamic scene handling.**  Under the pseudo-image representation of 3D priors, we can easily handle dynamic objects by accumulating 3D points of moving objects along their driving trajectory. This is a simple yet effective solution for handling dynamic objects, as we will demonstrate in response to Weakness #2 below.
> 2. **Training model on limited data.** To train the model synthesizing views beyond recorded trajectories without the respective ground truth data, we propose to formulate the generation process in an inpainting fashion and simulate camera movements with augmentation (as described in Sec. 3.1, lines 185-198) on pseudo-images. Both of these designs are effective in alleviating the shortage of training data in driving scenes.
> 3. **Cross-frame and cross-view information fusion.** By representing 3D scenes with sparse, colored 3D points, we can naturally aggregate information across frames and multi-views through point cloud accumulation, therefore ensuring the temporal and spatial consistency of generated results.  We will also demonstrate the strong temporal consistency of our proposed method later in response to weakness #2.
>
> As far as we know, FreeVS is the first NVS method to be able to render high-quality views far from captured views (e.g., when laterally moving the frontal camera for 4m), which we think is a major progress in driving scene NVS and cannot be trivially achieved.
>
> Reviewer 6Zxi recognized the novelty of our pseudo-image representation of 3D priors, while Reviewer YUkf acknowledged that the design and implementation of our generation pipeline are nontrivial and non-conventional.

---

> ### Author Response · Authors · 2024-11-17
> **Response to Reviewer BdtD (2/3)**
>
> **W2. About the temporal consistency of generated results and performance on dynamic objects**
>
> Our proposed generation pipeline can naturally leverage temporal information and generate temporally consistent results, and handle dynamic objects effectively.
>
> **Ensuring temporal consistency.** Being generated from accurate 3D points, pseudo-images are always consistent with real scenes, which ensures the fundamental temporal consistency performance of FreeVS.  Moreover, the pseudo-image representation can naturally offer 3D priors with high temporal consistency through accumulating 3D points across data frames (as described in lines 171-173 and lines 344-345).  Finally, we formulate the proposed pipeline as a video generation pipeline, which also strengthens the temporal consistency of generated videos.
>
> **Handling dynamic objects.** Dynamic objects will not cause trouble in generating pseudo-images. For each dynamic object, we accumulate its 3D points along its moving trajectory and project its 3D points onto each pseudo-image based on its current position. Therefore nearly no artifacts will be introduced by dynamic objects into pseudo-images. We think the easy and effective handling of dynamic objects is a unique advantage of the proposed pseudo-image representation.
>
> With the demo video submitted as supplementary material and Figure A in the appendix, we already show that FreeVS has strong performance on both temporal consistency and dynamic object handling.
>
> **To further prove this, we provide several long video sequences generated by FreeVS on an anonymized page, as posted at the beginning of this comment** (https://anonymize289.github.io/anonymized289.github.io/).  These videos showcase scenes with dynamic objects, clearly demonstrating FreeVS's ability to maintain temporal consistency and handle dynamic objects even with changes in trajectory and scene content.  FreeVS outperforms previous NVS methods and even many video generation works on temporal consistency and dynamic object handling.
>
> **W3: About generating views far from captured views.**
>
> **Yes, the pseudo-image representation surely has some limitations**. For example, the pseudo-image representation cannot handle non-rigid dynamic objects (like pedestrians and cyclists) as well as rigid dynamic objects since their 3D points are hard to be correctly accumulated across frames.
>
> However, thanks to the generalization ability of video generation models, views of pedestrians and cyclists synthesized by FreeVS are still satisfying as shown in the posted videos in response to W2, and Figure 5 in the paper.
>
> FreeVS also cannot perfectly recover scene contents when too few observations are provided.  However, FreeVS can still outperform previous NVS methods in such situations, as shown in Figure 4. We have added discussions on the limitation of the pseudo image representation in Sec. A.8 in the revised version of our paper.
>
> **As for synthesizing views far from captured views**, as reported in Table 2,  FreeVS has a greater performance advantage on views far from captured views.  Therefore synthesizing far views is clearly not a relative weakness of the proposed FreeVS. Indeed, pseudo-images can not be perfectly generated on views too far from the captured views when most scene contents in the novel views are not observed. However, such views are usually not considered in the NVS task since the NVS task focuses on generating novel views of the observed scene contents.
>
> As shown in the demo videos, FreeVS can synthesize high-quality images when laterally moving the ego vehicle for up to 5 meters, which is enough for simulating normal driving behaviors like lane changing.  Although FreeVS cannot synthesize viewpoints at arbitrary distances from the captured views,  its capable range of synthesizing high-quality views still far exceeds that of current approaches.

---

> ### Author Response · Authors · 2024-11-17
> **Response to Reviewer BdtD (3/3, References)**
>
> *References:*
>
> [1] Guo, J., Deng, N., Li, X., Bai, Y., Shi, B., Wang, C., ... & Li, Y. (2023). Streetsurf: Extending multi-view implicit surface reconstruction to street views. *arXiv preprint arXiv:2306.04988*.
>
> [2] Wu, Z., Liu, T., Luo, L., Zhong, Z., Chen, J., Xiao, H., ... & Zhao, H. (2023, July). Mars: An instance-aware, modular and realistic simulator for autonomous driving. In *CAAI International Conference on Artificial Intelligence* (pp. 3-15). Singapore: Springer Nature Singapore.
>
> [3] Xie, Z., Zhang, J., Li, W., Zhang, F., & Zhang, L. (2023). S-nerf: Neural radiance fields for street views. *arXiv preprint arXiv:2303.00749*.
>
> [4] Turki, H., Zhang, J. Y., Ferroni, F., & Ramanan, D. (2023). Suds: Scalable urban dynamic scenes. In *Proceedings of the IEEE/CVF Conference on Computer Vision and Pattern Recognition* (pp. 12375-12385).
>
> [5] Yang, J., Ivanovic, B., Litany, O., Weng, X., Kim, S. W., Li, B., ... & Wang, Y. (2023). Emernerf: Emergent spatial-temporal scene decomposition via self-supervision. *arXiv preprint arXiv:2311.02077*.
>
> [6] Chen, Y., Gu, C., Jiang, J., Zhu, X., & Zhang, L. (2023). Periodic vibration gaussian: Dynamic urban scene reconstruction and real-time rendering. *arXiv preprint arXiv:2311.18561*.
>
> [7] Tonderski, A., Lindström, C., Hess, G., Ljungbergh, W., Svensson, L., & Petersson, C. (2024). Neurad: Neural rendering for autonomous driving. In *Proceedings of the IEEE/CVF Conference on Computer Vision and Pattern Recognition* (pp. 14895-14904).
>
> [8] Zhou, X., Lin, Z., Shan, X., Wang, Y., Sun, D., & Yang, M. H. (2024). Drivinggaussian: Composite gaussian splatting for surrounding dynamic autonomous driving scenes. In *Proceedings of the IEEE/CVF Conference on Computer Vision and Pattern Recognition* (pp. 21634-21643).
>
> [9] Yan, Y., Lin, H., Zhou, C., Wang, W., Sun, H., Zhan, K., ... & Peng, S. (2024). Street gaussians for modeling dynamic urban scenes. *arXiv preprint arXiv:2401.01339*.

---

### Official Review · Reviewer_zGyV · 2024-11-02

**Soundness:** 3
**Presentation:** 3
**Contribution:** 3
**Rating:** 8
**Confidence:** 5

**Summary:**

The paper introduces FreeVS, an approach to view synthesis for driving scenes that overcomes limitations of existing methods, which primarily focus on synthesizing camera views along pre-recorded vehicle trajectories. Traditional methods tend to degrade in performance when synthesizing viewpoints deviate from the recorded trajectory, as these viewpoints lack of groundtruth data.

To ensure that the generated images remain consistent with the 3D structure of the scene and accurate in terms of viewpoint pose, the authors introduce a pseudo-image representation of view priors based on LiDAR. This representation controls the generation process and allows for the simulation of viewpoint transformations, enabling the model to mimic camera movement in different directions.

The authors proposed two benchmarks for evaluating novel camera synthesis and novel trajectory synthesis. The proposed method is evaluated on Waymo Open Dataset.

**Strengths:**

- The paper is well motivated, shows great potential for the real-application of autonomous driving simulation.
- Proposed a "psuedo LiDAR controlnet" for SVD, which is easy yet effective.
- The experimental results and demo video demonstrate the effectiveness of the proposed method.

**Weaknesses:**

- The evaluation is not very comprehensive. The baseline methods are not *specifically* designed for the similar purpose of the paper. 3D-GS is the basic method, EmerNeRF and StreetGaussian are more for the dynamic NeRF/3DGS. There are works that use virtual warping for improving the novel view quality such as [1] [2], that might be better for the baselines.
- The benchmark of novel trajectory synthesis looks interesting to me, however, the authors only show the FID results, while FID is well-known for its instability and unreliability by changing the resolutions, etc. This reminds me of the existing novel trajectory synthesis benchmark [3], the authors should test their methods on such a dataset and demonstrate the *absolute* performance gain using the metrics of PSNR, SSIM, etc.
- I would like to further know the zero-shot generalization of the trained SVD, since the Waymo Open dataset is quite clean for its camera/LiDAR extrinsic calibration, and the number of LiDAR beams is relatively high, it is interesting to show the results of genelizating it to other scenes or datasets, and how the SVD benefits the downstream 3DGS reconstruction.
- What are the results of applying the proposed methods on dynamic scenes?

[1] UC-NeRF: Neural Radiance Field for Under-Calibrated Multi-view Cameras in Autonomous Driving, ICLR 2024.

[2] HO-Gaussian: Hybrid Optimization of 3D Gaussian Splatting for Urban Scenes, ECCV 2024.

[3] XLD: A Cross-Lane Dataset for Benchmarking Novel Driving View Synthesis, arxiv, 2024.

**Questions:**

See above. I personally like the idea of the paper, but I still have many concerns and would provide a final rating based on the authors' responses.

---

> ### Author Response · Authors · 2024-11-21
> **Response to Reviewer zGyV (1/3)**
>
> **We sincerely thank you for your valuable suggestions. Here are our responses to your comments:**
>
> **W1. Comparing with methods with virtual warping:**
>
> Here we compare the performance of FreeVS and UC-NeRF on the validation sequences.
>
> Following the official implementation (https://github.com/kcheng1021/UC-NeRF) of UC-NeRF, we **only consider the three frontal cameras** in the following supplementary experiments, due to UC-NeRF’s mostly hard-coded implementations and hyper-parameters optimized for the 3-camera setting.
>
> We compare FreeVS with UC-NeRF under the novel-frame synthesis setting on the recorded trajectories and under the novel-trajectory synthesis setting. Considering dropping the side cameras will have some impact on the overall performance of each method, here we also provide the performance of EmerNerf under the 3-camera setting as a reference.
>
> For evaluating the perceptual robustness performance of each method, we only consider ground truth labels that are visible in the three frontal cameras.
>
> | Novel-frame synthesis | SSIM↑ | PSNR↑ | LPIPS↓ |
> | --- | --- | --- | --- |
> | EmerNerf | 0.764 | 25.16 | 0.282 |
> | UC-NeRF | **0.770** | **25.91** | 0.250 |
> | FreeVS | 0.761 | 25.47 | **0.146** |
>
> | Novel-trajectory synthesis  | FID↓ (y±1m) | LET-mAP↑ (y±1m) | FID↓ (y±2m) | LET-mAP↑ (y±2m) | FID↓ (y±4m) | LET-mAP↑ (y±4m) |
> | --- | --- | --- | --- | --- | --- | --- |
> | EmerNerf | 50.58 | 0.541 | 62.86 | 0.506 |  79.61 | 0.418 |
> | UC-NeRF | 42.08 | 0.592 | 75.56 | 0.447 | 88.47 |  0.370 |
> | FreeVS | **15.17** | **0.781** | **18.49** | **0.761** | **24.07** | **0.696** |
>
> On the recorded trajectories, UC-NeRF and FreeVS have similar performance (UC-NeRF has a slightly better SSIM/PSNR performance, while FreeVS has a significantly better LPIPS performance.) On the new trajectories, FreeVS significantly outperforms UC-NeRF.
>
> As for HO-Gaussian, we did not find its open-sourced implementation, making it very challenging to finish the respective experiments with HO-Gaussian before the rebuttal period ends.
>
> [1] XLD: A Cross-Lane Dataset for Benchmarking Novel Driving View Synthesis, arxiv, 2024.

---

> ### Author Response · Authors · 2024-11-21
> **Response to Reviewer zGyV (2/3)**
>
> **W2. Experiment on the XLD dataset:**
>
> Thanks for your constructive suggestion. We have managed to apply FreeVS on the XLD benchmark. We will quantitatively and qualitatively compare FreeVS with reconstruction-based methods (based on their performance reported in the XLD dataset paper[1]) on the XLD dataset below:
>
> First, we report the quantitative performance of FreeVS on the XLD dataset with different offsets, following the experiment setting of Table 1 in [1]:
>
> | Method | Offset 4m: PSNR↑ | Offset 4m: SSIM↑ | Offset 4m: LPIPS↓ | Offset 2m: PSNR↑ | Offset 2m: SSIM↑ | Offset 2m: LPIPS↓ |
> | --- | --- | --- | --- | --- | --- | --- |
> | UC-NeRF | 22.89 | 0.768 | 0.420 | 25.17 | 0.863 | 0.367 |
> | MARS | 23.29 | 0.818 | 0.235 | 24.95 | 0.847 | 0.194 |
> | EmerNeRF | 24.80 | 0.837 | 0.203 | 26.05 | 0.852 | 0.182 |
> | PVG | 23.17 | 0.841 | 0.353 | 24.42 | 0.854 | 0.335 |
> | FreeVS | **26.52** | **0.865** | **0.151** | **27.65** | **0.876** | **0.135** |
>
> | Method | Offset 1m: PSNR↑ | Offset 1m: SSIM↑ | Offset 1m: LPIPS↓ | Offset 0m: PSNR↑ | Offset 0m: SSIM↑ | Offset 0m: LPIPS↓ |
> | --- | --- | --- | --- | --- | --- | --- |
> | UC-NeRF | **30.07** | **0.896** | 0.355 | 35.95 | 0.936 | 0.311 |
> | MARS | 27.40 | 0.851 | 0.169 | 30.21 | 0.873 | 0.146 |
> | EmerNeRF | 28.66 | 0.878 | 0.150 | 31.76 | 0.907 | 0.126 |
> | PVG | 26.84 | 0.882 | 0.296 | **37.78** | **0.960** | 0.189 |
> | FreeVS | 29.08 | 0.894 | **0.120** | 30.06 | 0.908 | **0.079** |
>
> Experiments show that FreeVS outperforms all previous methods on all metrics on trajectories with 2m or 4m offsets.  On trajectories with 1m offsets, the performance of FreeVS is still better than most previous methods. On trajectories with no offset, reconstruction-based methods have better PSNR and SSIM performance. Still, we think the high performance of reconstruction-based methods on the original trajectories comes from their overfitting on the training views, considering the huge performance gap between their performance on trajectories with no offset and trajectories with a slight 1m offset (-5.88 PSNR for UC-NeRF,-10.94 PSNR for PVG).
>
> It is also worth emphasizing that FreeVS has a significantly better LPIPS performance on all validation trajectories, even including trajectories with no offsets. As a perceptual metric, LPIPS is more aligned with human perception compared with pixel-error metrics such as PSNR/SSIM.  As the performance degradation of the reconstruction-based methods mainly comes from artifacts in the image when facing out-of-domain test views, the performance degradation of FreeVS mainly comes from the loss of high-frequency details such as the brick patterns on a wall or the number of leaves on a tree.  We believe this is why FreeVS exhibits significantly better visual appeal in visualized results.
>
> We also present a qualitative comparison between FreeVS and reconstruction-based methods through an anonymized page: https://anonymize289no2.github.io/.   We compare the generation results of FreeVS with all visualization results shown in Figure 3 in the XLD dataset paper[1].
>
> [1] XLD: A Cross-Lane Dataset for Benchmarking Novel Driving View Synthesis, arxiv, 2024.

---

> ### Author Response · Authors · 2024-11-21
> **Response to Reviewer zGyV (3/3)**
>
> **W3. Zero-shoot generalization to other datasets / can FreeVS benefit the downstream reconstruction process**
>
> FreeVS models trained on the Waymo Open dataset (WOD) can be applied to any unseen sequences in the WOD dataset and generate high-quality results.
>
> However, like most current image or video generation models, FreeVS has limited adaptability to changes in the image domain like imaging styles and lighting conditions.  The current FreeVS models are also indeed sensitive to the number of LiDAR beams.
>
> We showcase two examples of applying the FreeVS model trained on the WOD dataset straight to the XLD or nuScenes dataset in a zero-shot manner on the anonymized page (https://anonymize289no2.github.io/).
>
> Like the WOD dataset, XLD dataset also has a high number of LiDAR beams, which leads to pseudo-images with dense valid pixels. Therefore FreeVS generalized to the XLD dataset can generate images with mostly correct scene contents. The generated result is mainly different from GT images in their image style.
>
> The nuScenes dataset has significantly fewer LiDAR beams compared with the WOD dataset. FreeVS applied on the nuScenes dataset has difficulties in recovering the scene contents. Training-time augmentations on LiDAR beams might alleviate this problem. However, we find it difficult to verify this during the rebuttal period.
>
> Finally, yes, we think the ability to generate out-of-trajectory novel views of FreeVS can benefit scene construction by providing extra pseudo observations. We plan to work on this in the future.
>
> **W4. About performance on dynamic scenes:**
>
> FreeVS has a very strong performance on dynamic scenes.  In fact, none of our validation segments are static scenes and more than half of them riches dynamic objects ( such as segment 1191788760630624072_3880_000_3900_000, segment 10444454289801298640_4360_000_4380_000, segment 10625026498155904401_200_000_220_000, segment 10963653239323173269_1924_000_1944_000, segment 12161824480686739258_1813_380_1833_380, and segment 18111897798871103675_320_000_340_000. )
>
> As described by lines 171-173 in the paper, for each dynamic object, we accumulate its 3D points along its moving trajectory and project its 3D points onto each pseudo-image based on its current position. Therefore nearly no artifacts will be introduced by dynamic objects into pseudo images. We think the easy and effective handling of dynamic objects is a unique advantage of the proposed pseudo-image representation.
>
> **Here we provide several long video sequences generated by FreeVS on another anonymized page**: https://anonymize289.github.io/anonymized289.github.io/.
>
> These videos showcase scenes with rich dynamic objects, demonstrating FreeVS's ability to handle dynamic objects well even with appearance or movement modifications on dynamic objects.

---

> > ### Comment · Reviewer_zGyV · 2024-11-22
> >
> > Thank you for providing the detailed response. I appreciate the authors' efforts devoted to the extra experiments with UC-NeRF and HO-Gaussian given the limited time. The authors should add those results and discussions to the main paper. The absolute comparison of the XLD dataset significantly strengthens the completeness of the proposed method, and I also suggest the authors incorporate those results, both quantitatively and qualitatively, into the paper.
> >
> > While most of my concerns have been well addressed by the authors, I have a few more questions:
> > - Regarding the dynamic scenes, the video results are impressive, but I want to understand how the authors generate those frames in practice. Do we need precise annotations such as bounding box and tracking results in 3D to warp those accumulated point clouds to novel views for diffusion?
> > - I understand that the rebuttal period might be too short for authors to train the diffusion models on nuScenes, would it be possible to add those results before the camera-ready version? It would be a shame that this paper is claimed to have good generalizable ability while missing those results.
> > - For the additional results on XLD dataset, why are only images shown on the webpage? Would itbe  possible to present videos even if they are short, otherwise, it is hard to evaluate the temporal and spatial consistency.

---

> > > ### Author Response · Authors · 2024-11-22
> > >
> > > **Thank you again for your valuable suggestions and prompt response.**
> > >
> > > **We have added the above-reported experiment results to the paper.**  As explained above, the current experiment setting of UC-NeRF (3 cameras) is not completely aligned with the experiment setting in the main paper (5 cameras).  We will modify the implementation of UC-NeRF and report its performance in the 5-camera setting in the camera-ready version.  For now, we temporarily include the above-reported experiment results in the appendix.
> > >
> > > Here are our responses to the new questions:
> > >
> > > - Yes, we need 3D bounding boxes of moving objects to accumulate and project their 3D points. In our experiment, we use ground truth annotations. We think predicted tracking results, especially tracking results optimized with the unicycle model can function as well, as demonstrated by previous works like HUGS[1].
> > >
> > > - We will try our best to generalize the proposed pipeline to the nuScenes dataset.  We found the sparse LiDAR beams in nuScenes dataset have indeed hindered our method's performance, and we will try solutions like adding pseudo-LiDAR points (generated from the predicted depth map) to solve this. We will report our experiment results on the nuScenes dataset in the camera-ready version.
> > >
> > > - Yes, of course. We have updated the posted page to add video results on the XLD dataset. (https://anonymize289no2.github.io/).
> > >
> > > [1] HUGS: Holistic Urban 3D Scene Understanding via Gaussian Splatting, CVPR2024

---

> > > > ### Comment · Reviewer_zGyV · 2024-11-23
> > > >
> > > > Thanks for the response. I think the authors well address my concerns. Therefore, I raise my score to accept.

---

### Official Review · Reviewer_NG25 · 2024-11-03

**Soundness:** 3
**Presentation:** 3
**Contribution:** 3
**Rating:** 6
**Confidence:** 3

**Summary:**

This work focus on novel view synthesis on extrpolated view (i.e. rendered views are far away from source training views).
The authors propose to use reprojected colored lidar points as condition, using a freezed diffusion decoder to achieve high quality synthesized images.
Experiments are conducted on Waymo, demonstrating way better results over other 3d-optimization based approaches (emernerf, streetgaussian)

**Strengths:**

1. Much better (more robust) results over 3D-optimization based approaches (EmerNerf, streetgaussian) on far-away novel veiws, because they typically have overfitting issues.
2. A combination of 3D informtion and 2D diffusion  model that provides both controllability and decent rendering results.

**Weaknesses:**

1. The rendering speed is very slow,  while 3DGS which can render at real time (50+fps), this hinder the downstream applications that requires realtime efficiency.
2. Inconsistency results because of using large decoder
3. Worse performance compared to 3D-optimization approaches if the novel view are close to source views;

**Questions:**

This idea is quite similar to Free View Synthesis which both project source views into target view, you may consider add it as a baseline, at least cite it.

@article{riegler2020free,
  title     = {Free View Synthesis},
  author    = {Gernot Riegler and V. Koltun},
  journal   = {European Conference on Computer Vision},
  year      = {2020},
  doi       = {10.1007/978-3-030-58529-7_37},
  bibSource = {Semantic Scholar https://www.semanticscholar.org/paper/49fae04a4e9383080788759f63dba75c86bd21b0}
}


You may also want to cite magicDrive and MagicDrive3D, as both work on generative driving scenes.

---

> ### Author Response · Authors · 2024-11-23
> **Response to Reviewer NG25 (1/2)**
>
> **We sincerely thank you for your valuable suggestions. Here are our responses to your comments:**
>
> **Q1: About missing citations / compare with Free View Synthesis**
>
> Thank you for reminding us of those missing citations. We have added citations to those works in the related works section (Sec. 2.2, lines 135, lines 146-147) in the revised version of our paper.
>
> As for comparing with Free View Synthesis (FVS), we have attempted to apply FVS to the WOD dataset but did not obtain satisfactory results. **We have shown one typical failure case of FVS when synthesizing views of open, dynamic scenes in the appendix of our paper (Sec A.11 and Figure E, page 20).**
>
> FVS is an early generation-based novel view synthesis method specially designed for narrow, static scenes, which are mostly object-centric. We found applying FVS on open, dynamic scenes in the WOD dataset usually can not generate high-quality views, as shown in Figure E.
> We believe this is due to FVS's optimization process relying on scene modeling with the COLMAP algorithm, which can not handle dynamic objects.
> FVS also does not consider dynamic objects in its pipeline.
> Additionally, the significantly more dispersed and sparse viewpoints in driving scenes also contribute to FVS's poor performance.
>
> In such a case, we think that FVS performs too weakly as a baseline in driving scenes. We have added citations to it as above-mentioned.

---

> > ### Author Response · Authors · 2024-11-23
> > **Response to Reviewer NG25 (2/2)**
> >
> > Moreover, we also would like to respond to the commented weakness of our method too:
> >
> > **W1. About the rendering speed.**
> >
> > Yes, if we ignore the scene reconstruction process (which usually takes 1-2 hours on a 20s sequence) of 3DGS-based methods, FreeVS currently runs slower than 3DGS when rendering novel views.
> >
> > But considering FreeVS can skip the time-consuming scene reconstruction process when applied on newly collected sequences, we believe that FreeVS has a unique application advantage over reconstruction-based methods.
> >
> > Considering such a situation that you want to synthesize novel views on an unseen new sequence, with FreeVS, you can simply load the model pre-trained on training sequences and inference on the new sequence. It only takes several minutes for FreeVS to generate novel views on an unseen sequence.  However, with reconstruction-based methods, you have to wait for 1-2 hours until the scene-reconstruction process is converged and then synthesize novel views. No preparations in advance can be made to mitigate this reconstruction cost.
> >
> > Finally, our proposed pipeline is decoupled with the specific generation model. We can expect the further speeding up of future video generation models, which is currently a trending research topic in the area of video generation.
> >
> > **W2. Inconsistency results  because of using large decoder**
> >
> > **We provide several long video sequences generated by FreeVS on an anonymized page: https://anonymize289.github.io/anonymized289.github.io/.**  As shown in the long videos, FreeVS has a satisfactory temporal consistency performance.
> >
> > The strong temporal consistency performance of FreeVS originates from several factors. Firstly, being generated from accurate 3D points, the proposed pseudo image priors are always consistent with real scenes, which ensures the fundamental temporal consistency performance of FreeVS.  Moreover, the pseudo-image representation can naturally offer 3D priors with high temporal consistency through accumulating 3D points across data frames (as described in lines 171-173 and lines 344-345).  Finally, we formulate the proposed pipeline as a video generation pipeline, which also strengthens the temporal consistency of generated videos.
> >
> > **W3. Performance when the novel views are close to the source views**
> >
> > As reported in Table 3, FreeVS outperforms previous methods by a large margin even in the recorded trajectory, when all five multi-view cameras are considered.  Since we sample test frames for each 4 frames following the conventional setting, the test views are very close to source views.
> >
> > Actually, the previous reconstruction-based method can only outperform FreeVS when only the frontal camera is considered, and their high performance is achieved through overfitting the reconstructed model on the very limited camera views.

---

> > > ### Comment · Reviewer_NG25 · 2024-11-25
> > > **thanks for the response**
> > >
> > > 1. FVS seems to be a decent baseline presented in https://arxiv.org/pdf/2308.01898. But I'm good with the status of FVS, tuning the FVS for driving scene requires significant work.
> > > 2. If you compare any dynamic vehicles in 1-2 second horizon, you can see difference especially in shape/geometry,  though the nearby frames are rather consistent.
> > > 3. About rendering speed. Many approaches can do quick/instant reconstruction and real-time rendering together,  I would suggest you to add more discussion on comparing with generalizable approaches that produce a standalone represenation that allows for real-time rendering in the revision, e.g.
> > >
> > > [1]  Chen, Yuedong, et al. "Mvsplat: Efficient 3d gaussian splatting from sparse multi-view images." European Conference on Computer Vision. Springer, Cham, 2025.
> > >
> > > [2]  Xu, Haofei, et al. "Depthsplat: Connecting gaussian splatting and depth." arXiv preprint arXiv:2410.13862 (2024).
> > >
> > > [3]  Chen, Yun, et al. "G3r: Gradient guided generalizable reconstruction." European Conference on Computer Vision. Springer, Cham, 2025.
> > >
> > > [4] Ren, Xuanchi, et al. "SCube: Instant Large-Scale Scene Reconstruction using VoxSplats." arXiv preprint arXiv:2410.20030 (2024).
> > >
> > > [5] Charatan, David, et al. "pixelsplat: 3d gaussian splats from image pairs for scalable generalizable 3d reconstruction." Proceedings of the IEEE/CVF Conference on Computer Vision and Pattern Recognition. 2024.

---

> > > > ### Author Response · Authors · 2024-11-26
> > > > **Thanks for the prompt response**
> > > >
> > > > **Thank you again for your valuable suggestions and prompt response.**
> > > >
> > > > Here are our responses to the new comments:
> > > >
> > > > **About the consistency of dynamic objects.**  We have updated the posted page ( **https://anonymize289.github.io/anonymized289.github.io/**) to showcase videos generated by FreeVS in the original trajectory. By comparing videos generated in the original trajectory and the modified trajectory, we can find that many object appearance inconsistencies (like the front vehicles in Scene 2) can be attributed to the significantly modified viewpoints.  We acknowledge that FreeVS still faces challenges when handling significant viewpoint changes. However, its robustness to viewpoint variations remains significantly stronger than that of existing methods. Also, constrained by the resolution of image generation, FreeVS sometimes can not recover all the appearance details of distant objects. Generating temporally consistent and high-resolution videos remains a major challenge for current video generation models. We look forward to the development of more advanced video generation models that can address these issues.  We hope our response can address your concerns.
> > > >
> > > > **About rendering speed.** We have **revised the paper to add a paragraph in the related works part, Sec. 2.1,** to introduce works in fast scene reconstruction and generalizable reconstruction.  We cited works for object-centric or narrow scenes and compared works for driving scenes (SCube[1], DrivingRecon[2], G3R[3])  with FreeVS.
> > > >
> > > > As feed-forward reconstruction methods, the quality of images rendered by SCube and DrivingRecon are not comparable to current per-scene reconstruction methods, based on their reported qualitative and quantitative results.   We do not consider the GAN postprocessing in SCube here, which needs per-scene training.  Based on results reported by DrivingRecon, previous feed-forward reconstruction methods (e.g., pixelSplat[4], MVSplat[5]) have even worse performance in driving scenes. Meanwhile, the image quality performance of FreeVS is comparable to current per-scene reconstruction methods.
> > > >
> > > > G3R has a very strong performance, but it only reported experiment results on the PandaSet[6] dataset and has not released its code yet. We promise to compare with G3R in the future if it is open-sourced.
> > > >
> > > > Moreover, we also added a quantitative efficiency comparison with generalizable reconstruction methods in **Sec A.12 in the appendix, page 21.**  We compare the efficiency of FreeVS on the WOD dataset with generalizable reconstruction methods on the PandaSet dataset according to the results reported by G3R[3]. Though the experiment settings are not aligned, we hope this can give a rough quantitative efficiency comparison between FreeVS and previous generalizable reconstruction methods.

---

> > > > > ### Author Response · Authors · 2024-11-26
> > > > > **References**
> > > > >
> > > > > [1] Ren, X., Lu, Y., Liang, H., Wu, Z., Ling, H., Chen, M., ... & Huang, J. (2024). SCube: Instant Large-Scale Scene Reconstruction using VoxSplats. *arXiv preprint arXiv:2410.20030*.
> > > > >
> > > > > [2] Anonymous. Drivingrecon: Large 4d gaussian reconstruction model for autonomous driving. In
> > > > > Submitted to The Thirteenth International Conference on Learning Representations, 2024. URL
> > > > > https://openreview.net/forum?id=0PcJAHbSmc. under review. 3
> > > > >
> > > > > [3] Chen, Y., Wang, J., Yang, Z., Manivasagam, S., & Urtasun, R. (2025). G3r: Gradient guided generalizable reconstruction. In *European Conference on Computer Vision* (pp. 305-323). Springer, Cham.
> > > > >
> > > > > [4] Charatan, D., Li, S. L., Tagliasacchi, A., & Sitzmann, V. (2024). pixelsplat: 3d gaussian splats from image pairs for scalable generalizable 3d reconstruction. In *Proceedings of the IEEE/CVF Conference on Computer Vision and Pattern Recognition* (pp. 19457-19467).
> > > > >
> > > > > [5] Chen, Y., Xu, H., Zheng, C., Zhuang, B., Pollefeys, M., Geiger, A., ... & Cai, J. (2025). Mvsplat: Efficient 3d gaussian splatting from sparse multi-view images. In *European Conference on Computer Vision* (pp. 370-386). Springer, Cham.
> > > > >
> > > > > [6] Xiao, P., Shao, Z., Hao, S., Zhang, Z., Chai, X., Jiao, J., ... & Yang, D. (2021, September). Pandaset: Advanced sensor suite dataset for autonomous driving. In *2021 IEEE International Intelligent Transportation Systems Conference (ITSC)* (pp. 3095-3101). IEEE.

---

### Official Review · Reviewer_YUkf · 2024-11-07

**Soundness:** 2
**Presentation:** 3
**Contribution:** 3
**Rating:** 6
**Confidence:** 4

**Summary:**

This paper approaches the task of novel view synthesis of outside-trajectory viewpoints on driving videos. It does so by training a conditional video diffusion model on outside-trajectory views created through projection of existing 3D point clouds. It evaluates generated images outside trajectories using off-the-shelf models. Compared to baselines StreetGaussian, EmerNeRF and 3D-GS, it shows superior results on novel camera synthesis, multi-view novel frame synthesis, and outstanding results on novel trajectory synthesis. Qualitative video results show a significant improvement over baselines.

**Strengths:**

Impressive results on a challenging task
- Results are clearly better than baselines in novel camera synthesis, multi-view novel frame synthesis, and significantly better in novel trajectory synthesis (FID drops by towards 75%)
- Qualitative comparisons clearly back these results
- Video results show impressive generation well outside input trajectory, while other methods have severe artifacts / fail entirely

Creative use of 3D and off-the-shelf models to enable a non-conventional setup
- Novel View Synthesis is so often limited to input trajectories. In the case of cars, this makes the task fairly straightforward and limited due to constraints on using positions connected to the car.
- Instead, this work approaches prediction several meters away from car trajectories. It does so by utilizing colored LiDAR across multiple views to create point clouds it can project into pseudo-images. This is a nontrivial trick to implement effectively. This work shows it can be useful for training a generative model!
- Evaluation is also tricky in the pseudo-image setting, but FID and 3D Detection mAP are suitable metrics; while of course qualitative results are most important.


***Post-Rebuttal Update*** I leave my score at 6. The reviewers did a good job of addressing my concerns and I feel the paper should be accepted as it offers good contributions in 3D and video generation to yield an effective method. See my response to the rebuttal for more detail.

**Weaknesses:**

Could use clearer argument for method leading to performance gain
- Numbers in the ablations table do not match that in comparisons to baselines. Why not?
- Ablations show little impact on performance. When the FID of this method is less than a third of that of baselines, surely more than 10% of performance can be explained by choices. For example, how does training data impact performance? What about pretraining or architecture? If these are important, it feels the architecture should be described in more detail.
- I infer a lot of the performance is coming from training data, yet the main paper has little information about this. How big is the train set in terms of sequences, if they are chosen from WOD?
- The alternative explanation I fear is most of the performance is coming from Stable Video Diffusion. It should be very clear how strong this baseline is without the proposed contributions

**Questions:**

See Weaknesses

---

> ### Author Response · Authors · 2024-11-21
> **Response to Reviewer YUkf (1/2)**
>
> **Q1. About model performance in the ablations table.**
>
> We train the SVD model for 20,000 iterations for ablation experiments in Table 4 to save training costs, as explained in line 417.
>
> The model is trained for 40,000 iterations when compared with SOTA methods in Table 1/2/3 (as explained in line 349).
>
> **Q2. About the impact of ablation study / pretraining / model architecture**
>
> **Firstly, FID alone cannot reliably reflect the model's performance in the novel view synthesis (NVS) task.**
>
> The FID metric can compare the overall image distribution between synthesized images and ground truth images, but it can not assess the 3D geometric accuracy of the synthesized images at all.  Theoretically, the model can achieve high FID performance as long as it generates realistic images,  no matter whether it follows the viewpoint instruction or not, or even no matter whether it generates images of the given scene or not.
>
> To fix this, we propose the perceptual robustness metric to evaluate the geometric accuracy of generated views. The ablation experiments show great impacts on the geometric accuracy performance of models. Such as under the ablation experiment setting (d) in Table 4, the model has a very poor geometric accuracy since it can not follow the given viewpoint instructions, as shown in Figure D in the appendix.
>
> **As for the model architecture**, we did not specifically optimize the model design of the Stable Video Diffusion model.  In fact, our pipeline is decoupled with the design of the generation model. We initialize the model from a Stable Diffusion model checkpoint, which is a common practice.  The only unusual module in our pipeline is the pseudo-image encoder, and as reported in Table 4 ((a) vs (e)), our pipeline is not sensitive to the design of the pseudo-image encoder as well.  We think the key design of our work is the pseudo-image representation of 3D priors, the formulation of recovering images from pseudo-images, and the camera movement simulation technique as a training time augmentation.
>
> We will open-source our code if the paper is accepted and welcome anyone to evaluate the robustness of our proposed pipeline regarding the design of the generative model.

---

> ### Author Response · Authors · 2024-11-21
> **Response to Reviewer YUkf (2/2)**
>
> **Q3.  About the selection of training data.**
>
> FreeVS is trained on the whole WOD training set, except for the selected validation sequences (i.e. all the 798-12 = 786 remaining sequences). We did not specifically choose sequences for model training.
>
> We did not test the impact of the scale of training data on FreeVS before, and here we report the performance of FreeVS model trained with 25% training data (we randomly choose 197 training sequences from the above-mentioned 786 sequences):
>
> | training data | SSIM↑（novel frame） | PSNR↑（novel frame） | LPIPS↓（novel frame） | FID↓ (y±2m)（novel trajectory） | LET-mAP↑ (y±2m) （novel trajectory） |
> | --- | --- | --- | --- | --- | --- |
> | 100% | 0.730 | 24.96 | 0.179 | 16.60 | 0.724 |
> | 25% | 0.726 | 24.49 | 0.172 | 16.72 | 0.719 |
>
> We report the performance of models under the multi-view novel frame synthesis setting and under the novel trajectory synthesis setting (y±2m).  As the experiment result shows, FreeVS is not demanding for the scale of training data.
>
>
> **Q4. About the performance of vanilla Stable Video Diffusion (SVD)**
>
> The vanilla stable video diffusion model is unable to handle the novel view synthesis(NVS) task.
>
> Generation models used for novel view synthesis (NVS) need to possess two fundamental capabilities: generating images of a real scene and **generating images from the specific given viewpoints**. Vanilla SVD models lack the ability to precisely control the viewpoints of generated images since they do not take viewpoint priors as input.
>
> For the NVS task, the target camera viewpoint prior is essential since we have to tell the diffusion model to generate images in which viewpoint.  We think the most trivial way to encode the target camera viewpoint prior is to provide the model with a camera pose transformation matrix from the reference viewpoint to the target viewpoint.
>
> Therefore we think the most trivial baseline in our experiment will be an SVD model with reference images and camera pose transformation matrices as priors, as the experiment setting (c) in Table 4. Dropping more priors based on experiment setting (c) in Table 4 would make the SVD model completely incapable of performing NVS.
>
> Still, here we provide the experiment result of an extra experiment setting (f) as a supplement to Table 4, in which only the reference images are fed as priors to the generative model (i.e. a vanilla SVD model):
>
> | Setting | View Priors | Encoders | SSIM↑（novel frame） | PSNR↑（novel frame） | LPIPS↓（novel frame） | FID↓ (y±2m)（novel trajectory） | LET-mAP↑ (y±2m) （novel trajectory） |
> | --- | --- | --- | --- | --- | --- | --- | --- |
> | (c) | Image+Pose | 2D-Conv+MLP | 0.613 | 19.86 | 0.288 | 21.25 | 0.013 |
> | (f) | Image              | 2D-Conv | 0.601 | 19.80 | 0.282 | 21.27 | 0.008 |
>
> As reported, a vanilla SVD  model with only reference images as prior performs similarly to a model trained under ablation setting (c). This is because as discussed in the current Sec. A.5 of the appendix, the generative model trained under ablation setting (c) nearly cannot follow the given viewpoint instruction. Therefore the performance of the model trained under ablation setting (c) is close to a vanilla SVD model. These models will only generate videos following the data distribution of training data. Specifically, they will only generate videos in which the ego vehicle moves along the recorded trajectory, as shown in the current Figure C.  These models can not satisfy our requirements to generate views out of the recorded trajectories.

---

> > ### Comment · Reviewer_YUkf · 2024-11-22
> > **Rebuttal Response**
> >
> > Thanks for the authors for their thoughtful rebuttal. I think the SVD image-only baseline is helpful to understand performance. Although it is still strong, contributions do improve upon this in a nontrivial manner, making the final method highly competitive. Thanks for the additional experiments on training data and clarification on baseline numbers.
> >
> > I disagree with Reviewer BdtD's view temporal information is not considered, as outlined in the author rebuttal. Taking into account these comments and author responses, I therefore agree with the other reviewers this paper should be accepted. It has creative use of 3D and off-the-shelf models to enable a non-conventional setup leading to impressive results.

---

> > > ### Author Response · Authors · 2024-12-04
> > > **Thanks for Your Acknowledgment**
> > >
> > > We appreciate your acknowledgment of our work.  Considering that Reviewer BdtD did not engage in the rebuttal process to allow us to address their misunderstanding of our work, we currently have an average review rating of 5.8 (8/6/6/6/3), which might lead to a rejection of this work (Top 30.5% in total submissions, while the acceptance rate of ICLR is usually less than 30%.) Therefore, we would be most grateful if you could kindly consider raising your rating score for our work if you support its acceptance.

---

### Official Review · Reviewer_6Zxi · 2024-11-08

**Soundness:** 3
**Presentation:** 3
**Contribution:** 3
**Rating:** 6
**Confidence:** 3

**Summary:**

The authors present FreeVS -- a Video Stable Diffusion-based generative view synthesis method for driving scenes that can synthesize high-quality camera views both on and beyond recorded trajectories.

The key innovation is using pseudo-images created by projecting colored point clouds as a unified representation for view priors. As opposed to recent contenders that rely on gaussian splatting or nerfs to represent the scene, the authors train a diffusion model on colored LiDAR point clouds.

The authors introduce two new benchmarks to evaluate the generated images that are far from the original poses. The method outperforms previous approaches for both traditional and newly proposed benchmarks on the Waymo dataset.

**Strengths:**

### Novelty
Clever use of pseudo-images obtained through colored point cloud projection as a unified representation for all view priors, simplifying the learning objective for the generative model.
### Evaluation
Introduces two new challenging benchmarks - novel camera synthesis and novel trajectory synthesis.

### Efficiency
The authors claim it takes less computational resources at inference time compared to splatting-based models.

### Performance
Better performance versus contenting methods, especially on poses far away from the original camera poses.

**Weaknesses:**

### Novelty
Engineering work -- it boils down to an addon for Video Stable Diffusion that has colored LiDAR point features concatenated.


### Different type of artfacts
The method trades the gaussian and nerf artifacts with the diffusion ones. While there is no denying that FreeVS works better than the previous attempts from novel views, for single front view, splatting still yields significantly better results (Table 3, front view).

### Evaluation
A single dataset is benchmarked (Waymo Open Dataset).

### Paper quality
Some tables missing numbers -- Table 3, reconstruction time.

**Questions:**

1. Why is there no ablation study for **no priors**? This should be as close as possible to the vanilla Video Stable Diffusion.
2. Have you experimented with LiDAR noise / pseudo lidar from MDE methods or maybe a mesh-based method? Otherwise this pipeline is bound to an expensive data acquisition pipeline.
3. Training/inference times are unclear. How can it be that your method is faster than splatting-based methods at 0.9 FPS? The training time should also be discussed but it's not.
4. Could moving objects benefit from a special treatment? I.e, tracking/ inpainting. It doesn't look like the method handles well uncertain areas.

---

> ### Author Response · Authors · 2024-11-23
> **Response to Reviewer 6Zxi (1/2)**
>
> **We sincerely thank you for your valuable suggestions. Here are our responses to your questions:**
>
> **Q1. Generation with no priors.**
>
> **The vanilla stable video diffusion model with no prior inputs is unable to handle the novel view synthesis(NVS) task.**
>
> Generation models used for novel view synthesis (NVS) need to possess two fundamental capabilities: generating images of a real scene and **generating images from the specific given viewpoints**.
>
> We regard reference images as a default input prior, like most Stable Video Diffusion (SVD) models for driving scenes.
>
> The target camera viewpoint is also an essential condition since we have to tell the diffusion model to generate images in which viewpoint.  We think the most trivial way to encode the target camera viewpoint prior is to provide the model with a camera pose transformation matrix from the reference viewpoint to the target viewpoint.
>
> Therefore we think the most trivial baseline in our experiment will be a video generation model with reference images and camera pose transformation matrices as priors, as the experiment setting (c) in Table 4.
>
> Dropping more priors based on experiment setting (c) in Table 4 would make the video diffusion model completely incapable of performing NVS.  Still, here we provide the experiment result of an extra experiment setting (f) as a supplement to Table 4, in which only the reference images are fed as priors to the generative model (i.e. a vanilla SVD model):
>
> | Setting | View Priors | Encoders | SSIM↑（novel frame） | PSNR↑（novel frame） | LPIPS↓（novel frame） | FID↓ (y±2m)（novel trajectory） | LET-mAP↑ (y±2m) （novel trajectory） |
> | --- | --- | --- | --- | --- | --- | --- | --- |
> | (a) | Pseudo images | 2D-Conv | 0.704 | 23.28 | 0.203 | 21.27 | 0.690 |
> | (c) | Image+Pose | 2D-Conv+MLP | 0.613 | 19.86 | 0.288 | 21.25 | 0.013 |
> | (f) | Image | 2D-Conv | 0.601 | 19.80 | 0.282 | 21.27 | 0.008 |
>
> As reported, a vanilla SVD  model with only reference images as prior performs similarly to a model trained under ablation setting (c). This is because as discussed in the current Sec. A.5 of the appendix, the generative model trained under ablation setting (c) nearly cannot follow the given viewpoint instruction. The performance of the model trained under ablation setting (c) is close to a vanilla SVD model. These models **can not precisely control the viewpoint of generated views**, and therefore can not handle the NVS task well.
>
>
> **Q2. Generation without ground truth LiDAR points.**
>
> **Yes, the LiDAR inputs for FreeVS can be replaced by pseudo-LiDAR points generated from estimated depths.**
>
> We experimented with applying an off-the-shelf Depth-Anything-V2[1] model on reference images and sample pseudo LiDAR points from the obtained depth map.  Here we show a success case as well as a failure case of generation with pseudo LiDAR points through an **anonymized page**: https://anonymize289no3.github.io/. We also provide a video comparison between results generated with GT LiDAR points or pseudo LiDAR points.
>
> According to our observation, the proposed FreeVS pipeline can be applied to pseudo LiDAR points and recover most scene contents correctly. However, we found the depth prediction model sometime has problems in predicting the depth of objects very near to the camera, as shown in the failure case, circled with red.  We believe this issue is caused by directly applying the Depth-Anything-V2 model to the WOD dataset. A depth predictor that is fully converged on the WOD dataset should be able to avoid this problem.  As for FreeVS, the visualized results show that our method has satisfying performance on most scene contents with about the right depth.
>
> Note that we generate the above-provided results by directly applying the FreeVS model trained with GT LiDAR points on the pseudo LiDAR points. The different valid pixel patterns and pixel density of pseudo image generated from pseudo LiDAR points lead to some slight imaging style changes of the generated result, which is totally avoidable by training FreeVS on pseudo image generated from pseudo LiDAR points. However, we do not have enough time to validate this during the rebuttal period.  We believe the above-provided results can prove that the proposed FreeVS does not heavily rely on GT depth. We have added the above-reported results to Sec. A.8 in the appendix.
>
> [1] Yang, L., Kang, B., Huang, Z., Zhao, Z., Xu, X., Feng, J., & Zhao, H. (2024). Depth Anything V2. *arXiv preprint arXiv:2406.09414*.

---

> > ### Author Response · Authors · 2024-11-23
> > **Response to Reviewer 6Zxi (2/2)**
> >
> > **Q3. Training cost discussion.**
> >
> > A key feature of FreeVS is that once the generative model is trained on training sequences, we can directly apply FreeVS to any newly collected driving sequences and generate views on novel viewpoints immediately.
> >
> > The model training of FreeVS takes 1-2 days on 8 L20 GPUs,  but you can train FreeVS on training sequences and save the trained model, then perform quick inference on newly collected sequences that share the same domain with the training sequences (e.g. validation sequences in the same dataset). You do not have to train FreeVS again whenever applying it to new sequences.
> >
> > In contrast, the reconstruction-based NVS pipelines can not be “pre-trained”.  They need to perform scene reconstruction on any newly collected driving sequences first (which usually takes 1-2 hours) and then synthesize novel views.
> >
> > Considering such a situation that you want to synthesize novel views on an unseen new sequence, with FreeVS, you can simply load the pre-trained model and inference on the new sequence. **It only takes several minutes for FreeVS to generate novel views on an unseen sequence.  However, with reconstruction-based methods, you have to wait for 1-2 hours until the scene-reconstruction process is converged and then synthesize novel views.** No preparations in advance can be made to mitigate this reconstruction cost.
> >
> > Therefore although the rendering speed of FreeVS is indeed inferior to 3DGS, the feature of not requiring a scene-wise reconstruction process still gives FreeVS a unique computational cost advantage. Therefore the "reconstruction time" of FreeVS is not filled in Table 3, which is not a typo, since FreeVS does not need a reconstruction process.
> >
> > Finally, our proposed pipeline is decoupled with the specific generation model. We can expect the further speeding up of future video generation models, which is currently a trending research topic in the area of video generation.
> >
> > **Q4. About handling moving objects**
> >
> > Yes, we found it very effective to use the bounding box trajectories of moving objects to handle dynamic objects.
> >
> >  As described by lines 171-173 in the paper, for each dynamic object, we accumulate its 3D points along its moving trajectory and project its 3D points onto each pseudo-image based on its current position. Therefore nearly no artifacts will be introduced by dynamic objects into pseudo images.  We think the easy and effective handling of dynamic objects is a unique advantage of the proposed pseudo-image representation.
> >
> > Here we provide several long video sequences generated by FreeVS on an anonymized page: https://anonymize289.github.io/anonymized289.github.io/.
> >
> > These videos showcase scenes with rich dynamic objects, demonstrating FreeVS's ability to handle dynamic objects well even with appearance or movement modifications on dynamic objects. Also, these videos show that FreeVS can handle most unobserved uncertain areas in novel camera views.   The pseudo image / generated image pairs shown in https://anonymize289no3.github.io/ can also demonstrate this, as the unobserved uncertain areas in the pseudo image are reasonably filled in the generated results.

---

### Meta-Review · Area_Chair_KyF9 · 2024-12-19

**Metareview:**

This paper presents a video diffusion model conditioned on projected colored LIDAR point clouds, and show that it enables high-fidelity novel view synthesis in autonomous driving scenarios. In particular, compared to prior methods like StreetGaussians which show results from viewpoints limited to driving trajectories, this approach allows rendering from different viewpoints. The other reviewers agree that the empirical results are promising and that the technical approach is sound and well-implemented. The AC agrees with this analysis, and feels this paper would be useful for the community, particularly in spurring view synthesis beyond trajectory viewpoints.

**Additional Comments On Reviewer Discussion:**

While this paper received mixed scores, the negative reviewer did not engage in discussion and did not cite specific methods for the novelty concerns raised. The AC is thus down-weighting the negative rating.

---

### Decision · Program_Chairs · 2025-01-22

Accept (Poster)